# Substitutive Effects of Milk vs. Vegetable Milk on the Human Gut Microbiota and Implications for Human Health

**DOI:** 10.3390/nu16183108

**Published:** 2024-09-14

**Authors:** Alicia del Carmen Mondragon Portocarrero, Aroa Lopez-Santamarina, Patricia Regal Lopez, Israel Samuel Ibarra Ortega, Hatice Duman, Sercan Karav, Jose Manuel Miranda

**Affiliations:** 1Laboratorio de Higiene Inspección y Control de Alimentos, Departamento de Química Analítica, Nutrición y Bromatología, Campus Terra, Universidade de Santiago de Compostela, 27002 Lugo, Spain; alicia.mondragon@usc.es (A.d.C.M.P.); aroa.lopez.santamarina@usc.es (A.L.-S.); patricia.regal@usc.es (P.R.L.); 2Área Académica de Química, Universidad Autónoma del Estado de Hidalgo, Carretera Pachuca-Tulancingo km. 4.5, Pachuca 42076, Hidalgo, Mexico; israel_ibarra@uaeh.edu.mx; 3Department of Molecular Biology and Genetics, Çanakkale Onsekiz Mart University, Çanakkale 17000, Türkiye; hatice.duman@comu.edu.tr (H.D.); sercankarav@comu.edu.tr (S.K.)

**Keywords:** cow milk, camel milk, gut microbiota, vegetable beverages, milk oligosaccharides, milk fat globule membrane

## Abstract

**Background:** In the last two decades, the consumption of plant-based dairy substitutes in place of animal-based milk has increased in different geographic regions of the world. Dairy substitutes of vegetable origin have a quantitative composition of macronutrients such as animal milk, although the composition of carbohydrates, proteins and fats, as well as bioactive components, is completely different from that of animal milk. Many milk components have been shown to have relevant effects on the intestinal microbiota. **Methods:** Therefore, the aim of this review is to compare the effects obtained by previous works on the composition of the gut microbiota after the ingestion of animal milk and/or vegetable beverages. **Results:** In general, the results obtained in the included studies were very positive for animal milk intake. Thus, we found an increase in gut microbiota richness and diversity, increase in the production of short-chain fatty acids, and beneficial microbes such as *Bifidobacterium*, lactobacilli, *Akkermansia*, *Lachnospiraceae* or *Blautia*. In other cases, we found a significant decrease in potential harmful bacteria such as Proteobacteria, *Erysipelotrichaceae*, *Desulfovibrionaceae* or *Clostridium perfingens* after animal-origin milk intake. Vegetable beverages have also generally produced positive results in the gut microbiota such as the increase in the relative presence of lactobacilli, *Bifidobacterium* or *Blautia*. However, we also found some potential negative results, such as increases in the presence of potential pathogens such as *Enterobacteriaceae*, *Salmonella* and *Fusobacterium*. **Conclusions:** From the perspective of their effects on the intestinal microbiota, milks of animal origin appear to be more beneficial for human health than their vegetable substitutes. These different effects on the intestinal microbiota should be considered in those cases where the replacement of animal milks by vegetable substitutes is recommended.

## 1. Introduction

Milk is defined as lacteal secretion from one or more healthy milch animals [1]. It is the first food in the diet of mammals, providing all the energy and nutrients necessary for growth and development in their first periods of life, when it is the main source of nutrition for infants until they can consume other foods [2]. In infants, milk is essential, especially human milk, which is rich in protein, fat, lactose, and a large variety of vitamins. Owing to its immense nutritional value, it has arguably been called “nature’s nearly most perfect food” [3]. After weaning, milk intake stops in all mammals except for humans, who continue their consumption in adulthood, not only as milk but also as dairy products [4]. However, because of their high nutritional value, milk and dairy products are regarded as staples of Western diets [4], are frequently included as important elements in a healthy and balanced diet and can provide the necessary energy and nutrients [5]. Among its nutrients, milk and dairy products contain high concentrations of micronutrients and macronutrients, such as calcium, magnesium, selenium, zinc, vitamin B_12_ and pantothenic acid, which are rich nutrients in the human diet [6]. Generally, a total of 3–4 serves/day of dairy products is recommended for adults, although this amount may vary depending on age, sex, and other physical requirements [7].

Many epidemiologic studies have confirmed the nutritional importance of milk in the human diet and reinforced the possible role of its consumption in preventing several chronic conditions, such as cardiovascular diseases, osteoporosis, some forms of cancer, obesity, and diabetes [5,6]. Moreover, milk also contains a multitude of proteins with anti-inflammatory properties, and these bioactive factors may attenuate intestinal inflammation [5]. Dairy products are also the primary source of dietary calcium. Recent studies have demonstrated that calcium intake is very low in many parts of the world and that the average daily intake of calcium per person is very low, under 400 mg/day [8]. In addition to its well-known relevance for the skeletal system, the importance of calcium intake extends well beyond its structural role because calcium is a pivotal player in various biochemical reactions and physiological processes, acting as a signal in many cellular processes [9]. Indeed, observational studies indicate that an inverse correlation exists between the intake of calcium and body weight [10].

However, despite the broad spectrum of benefits of milk and dairy food intake, the consumption of milk in some countries is decreasing [11]. Allergenicity associated with milk components, socioreligious beliefs about the consumption of animal products, disease phobia and the philosophical and ethical practice of veganism are some of the societal concerns related to animal-based milk. These concerns have led to an increased demand for nondairy alternatives [1]. Indeed, milk consumption is being replaced in some consumer groups by plant-based milk substitutes, which are presented to consumers as healthier, more sustainable, and animal-friendly alternatives to bovine milk [4].

Another important reason why many people give up or reduce milk consumption is the increase in the proportion of people, especially children, who are allergic to milk proteins. Approximately 2 to 7.5% of children suffer from cow milk (CM) protein allergy during their first year of life, which manifests as allergic symptoms in the skin, gastrointestinal tract, and respiratory tract [11]. An important proportion of the adult population in different countries is unable to completely digest lactose and is shelf-perceived as lactose intolerant [4]. For all those people who do not consume milk or dairy foods, plant-based formulas are among the alternatives available where the protein source is replaced by a plant-based protein. However, there may be some disadvantages regarding soy formula due to its potential effects on sexual development and reproduction, neurobehavioral development, immune function, and thyroid function [11].

Plant-based substitutes can be defined as an emulsion that resembles animal-origin milk in consistency and appearance, which are made following a general procedure that includes the aqueous extraction of the plant material, removal of remaining soil parts, and afterwards a thermal treatment of the fluid [12]. The most employed vegetable sourced to make plant-based beverages can be legumes, cereals, pseudocereals, seeds or nuts, oilseeds plants or even potatoes [13]. Nowadays, the most popular plant-based beverages are soy, almond, coconut, oat and rice [13].

In recent years, many articles evaluating the effects of the consumption of milk with probiotics or prebiotics on the human GM have been published [14,15,16,17,18]. Similarly, vegetable beverages supplemented with probiotics [19,20,21] or components from milk, such as dairy proteins [22,23], dairy fats [24,25,26], or vegetable proteins from vegetable beverages [27,28,29], were also recently published.

However, despite its common consumption, the number of articles that have investigated the effects of animal milk and its natural vegetable substitutes on the GM is relatively low. Thus, the aim of this narrative literature search was conducted up to June 2024 for all the available literature in the Web of Science and Scopus. A combination of the following search terms was applied: “milk”; “milk substitute”; “vegetal beverages”; “plant-based milk” as the topic; and “gut microbiota” in the title. A total of 64 articles were ultimately selected and included in the review, as were the other 48 articles that were included to contextualize the article.

## 2. The Human Gut Microbiota

The term “microbiota” refers to the entire microbial community (including commensal, mutualistic or pathogenic bacteria) that inhabits a given habitat; in this sense, “gut microbiota” (GM) refers to the range of microorganisms (bacteria, fungi, viruses, eukaryotes, *archaea*, and phages) that reside in the gastrointestinal tract [22]. Indeed, the adult GM is characterized by a specific presence and functionality of microbial species that are more favored in the gut environment [22,30]. The composition of the healthy gut microbiota is dominated for up to 90% by the phyla Firmicutes and Bacteroidetes [5]. The phylum of Firmicutes includes several genera, of which the most common (up to 95%) are *Lactobacillus*, *Bacillus*, *Enterococcus*, *Ruminicoccus* and *Clostridium* [31]. The GM plays key roles for human health, including a shield effect with protection of the gut barrier, shaping and maturation of the immune system, the regulation of human metabolism and nutrients and drug absorption [31].

The GM is a highly dynamic ecosystem influenced by numerous factors such as age, genetics, environment, lifestyle and, most prominently, diet. While different compositions of diet can lead to changes of microbiota profiles, an unbalanced diet consumed over a long period, such as the Western diet [32], accompanied by an unhealthy lifestyle can leave traces of evidence on the GM as well. Generally, subjects with obesity and overweight tend to show a lower diversity of GM [32].

In recent years, the GM has been extensively studied and explored because of its strong relevance to diseases, such as intestinal cancer, inflammatory bowel diseases, diabetes, metabolic syndrome, obesity and even brain diseases [33]. Studies have shown that the GM can not only influence the development and function of organ systems for adaptation and evolution but also protect them from exogenous microorganisms and toxins [6], aiding in the prevention or treatment of certain gastrointestinal disorders and promoting healthy and balanced microbial development in infants [34].

Complex cooperative relationships of coadaptation, coevolution, and interdependence exist between humans and the GM. The digestive system is the most inhabited section, but the degree of colonization is not identical. The diversity in the GM composition is attributed to environmental variations in the digestive tract. In addition, the GM has a valuable role in preserving health, principally contributing to the enhancement of immunity and controlling numerous basic metabolic routes [35]. In fact, changes in the GM impair homeostasis, causing GM-related diseases, such as gastrointestinal tract functional diseases, inflammatory bowel diseases, infectious intestinal diseases, gastrointestinal cancers, liver diseases, metabolic and obesity syndromes, allergies and diabetes [35] (Figure 1). In addition, neurological and neuropsychiatric disorders, including Alzheimer’s disease, major depression and autism, occur through the impairment of gut–brain axis homeostasis [22].

The metabolism of the GM results in the production of beneficial metabolites such short-chain fatty acids (SCFAs), which can be defined as the primary end products formed by non-digestible carbohydrates (in some cases also peptides and proteins) when fermented by GM [36]. The most important SCFAs are acetate, propionate and butyrate [13], which are easily soluble and absorbed quickly in >95% total, and the remaining are excreted in feces [36]. To produce each of them, a different mechanism and fermenting bacteria are needed. Acetate is mainly produced from pyruvate via acetyl CoA, requiring the participation of GM bacterial genus such as *Akkermansia*, *Bacteroides* or *Bifidobacterium* [36]. Propionate is formed from different synthesis ways: it can be produced from lactate (participating *Megaspharea elsdenii* and *Coprococcus catus*), from succinate (participating *Bacteroides*, *Dialister* or *Roseburia* genera) or from fucose and rhamnose (participating *Salmonella* and *Roseburia* genera) [37]. SCFAs are absorbed by colonocytes (mainly butyrate, a vital energy source for colonocytes) and effluxes into the blood [36]. SCFAs can also act as signaling molecules by activating G-protein-coupled receptors and inhibiting histone deacetylases [36], modulating gut health and immune processes, hormone synthesis, and lipogenesis [30].

They also participate in numerous interactions with the host [30]. Acetate, butyrate and propionate induce enteroendocrine L cells to release the YY peptide and glucagon-like neuropeptide-1 (GLP-1) [30]. These peptides regulate digestion and modulate lipid metabolism, which indirectly affects fatty acid storage in the liver [30]. Butyrate stimulates the intestinal epithelium, promotes GLP-1 release, and increases mucus secretion, reducing intestinal barrier permeability. It also has anti-inflammatory properties and protects against colitis and colon cancer.

The GM community can be affected by multiple factors, including diet, age, host species, and gastrointestinal tract parts. Nevertheless, diet and host species are the key factors contributing to the composition of the GM [38]. Evidence from animal studies shows that components of milk (such as fat and protein) and dairy derivatives (casein and whey) can prompt compositional changes to the GM, while there is some limited evidence in humans to show the impacts of some dairy groups on the GM [7].

## 3. Effect of Animal-Based Milk on the Gut Microbiota

Owing to its content of bioactive molecules, including proteins, lipids, and oligosaccharides, animal-based milk is considered a functional food whose antimicrobial, immunological, and antitumorigenic activities have been widely studied and exploited in the nutraceutical and biomedical fields to support the treatment and prevention of diseases [2,22]. A recent review [7] showed that milk and dairy products intake can modulate GM in a beneficial manner. Dairy foods appeared to facilitate the growth of beneficial microbes as lactobacilli and *Bifidobacterium*. However, due to the heterogeneity in study methods and outcome reporting, it is difficult to draw robust conclusions.

Among milk components, several authors have reported that several milk components can affect the GM. Among them, whey proteins [22], lactose [7,39], oligosaccharides [22], fatty components [25,40], and bioactive components such as lactoferrin [22,41] have been mentioned.

Lactose is the key milk carbohydrate and is often referred to as a “milk sugar” [7]. It is the predominant soluble digestible glycan in milk, serving primarily as a readily available energy source for newborn mammals [27]. Lactose intolerance depends on the absence of a gene named *LCT*, which provides instructions to produce an enzyme called lactase, which helps to digest lactose [39]. The *LCT* variant is associated with obesity and is regulated by lactose and milk consumption [42]. Previous studies have demonstrated that lactose intake increases the growth of lactobacilli and *Bifidobacterium*, which are correlated with health-related microbiota taxa [24], and that lactose also has a prebiotic index of 5.75, which is like that of many other prebiotics [7].

Milk oligosaccharides (MOs) are carbohydrates composed of 3–20 monosaccharide moieties [43] and are considered key growth factors for beneficial bacterial species [25]. Two main mechanisms have been associated with the modulation of the GM by MOs [22]. The first mechanism is related to the direct prebiotic activity of MOs, which selectively promote the growth of certain *Bifidobacterium* strains, including *Bifidobacterium infantis*, *Bifidobacterium brevis*, and *Bifidobacterium bifidum*, as well as some lactobacilli, such as *Lactobacillus bifidus* [22]. The second mechanism is related to the inhibition of colonization by various enteric pathogens, including *Vibrio cholerae*, *Salmonella fyris*, *Campylobacter jejuni*, *Clostridium difficile*, and various *Escherichia coli* strains, by preventing epithelial adhesion [22].

Whey proteins have several functions, including antioxidant, anticancer, antimicrobial, anti-inflammatory, and immunomodulatory effects. Furthermore, evidence suggests a beneficial role of whey proteins in positively modulating the GM in both infants and adults [22]. Among whey proteins, α-lactalbumin is the most prevalent in human milk, constituting approximately 35%, whereas in other animal milks, such as bovine milk, it is present in lower amounts, representing the second most abundant whey protein (approximately 17%) after *β*-lactoglobulin [22]. *β*-Lactoglobulin exhibits antimicrobial characteristics by inhibiting bacterial adhesion to the host surface and impeding pathogen colonization [44,45]. The antibacterial action of this substance is effective against several Gram-positive and Gram-negative bacteria, including *Bacillus subtilis*, *Staphylococcus aureus*, *E. coli*, and *Bordetella bronchiseptica* [45]. Importantly, α-lactalbumin and some of its bioactive peptides have been shown to play a role in the modulation of the GM by acting as prebiotics, and studies suggest that this effect is related both to the direct promotion of the growth of probiotics, including bifidobacteria, and, indirectly, to their antimicrobial activity against diverse pathogens [22]. For example, peptides derived from α-lactalbumin exhibit significant antimicrobial activity against many bacteria, including *Staphylococcus epidermidis* ATCC 12228, *Staphylococcus lentus*, and *B. subtilis BGA* [46]. Two components of milk whey proteins that exert relevant effects on gut bacterial species are lactoferrin and lysozyme [22].

Lactoferrin is a versatile glycoprotein that is present in the milk of the majority of mammals [47]. Lactoferrin plays an important role in promoting and maintaining the functional GM and inhibiting gut barrier impairment due to its strong antimicrobial activity, including its bacteriostatic and bactericidal effects, as well as its immunomodulatory functions, which help to preserve the integrity of the gut barrier [22]. It can attach to iron and engage with several pathogens, thus serving as the primary protective barrier in the mucous membranes of the body. Lactoferrin functions as a discerning antimicrobial agent by eliminating harmful microorganisms while promoting the growth of healthy microorganisms such as Lactobacilli and *Bifidobacterium* [48]. Petschow et al. reported that bovine lactoferrin selectively enhances the proliferation of *B. infantis* and *B. brevis*, whereas human lactoferrin significantly increases the growth of *B. infantis* [49]. Various in vitro studies have suggested that lactoferrin and its peptide derivative, lactoferricin, have prebiotic effects on the GM [41].

Finally, the antibacterial activity of the lysozyme has a role in reducing the complexity of the GM, increasing the resistance to intestinal colonization by some bacterial species, including pathogens, still favoring the growth of beneficial bacteria, and enhancing recovery from diverse gastrointestinal pathological conditions [50]. The mechanisms underlying the resistance of some probiotic strains to the antibacterial action of lysozyme have not yet been completely elucidated. However, the results from in vitro studies on human-residential bifidobacteria indicated that the tolerance of some bifidobacterial strains is attributable to the nonenzymatic antibacterial activity of lysozyme [22]. On the other hand, lysozyme exerts an antimicrobial effect by hydrolyzing *β*-1,4 glycosidic linkages in peptidoglycan, which is a structural component that determines the cellular shape and provides protection against turgor pressure. This enzymatic activity ultimately results in cell lysis and death [51].

Mammary epithelial cells secrete milk fat globules, which are surrounded by an intricate membrane known as the milk fat globule membrane (MFGM). A recent study indicated that MFGM proteins have beneficial effects, including promoting the development of a healthy gut microbiota and providing defense against infection and inflammation. The thin trilayer structure of MFGM consists of polar lipids such as phospholipids and sphingolipids and membrane proteins such as glycoproteins and enzymes [52,53]. In piglets, supplementation with bovine milk fat and milk fat globule membrane (MFGM), a membrane surrounding the fat globules in milk [40], increased the abundances of the Proteobacteria and Bacteroidetes phyla while decreasing the abundance of the Firmicutes phylum compared with those in piglets receiving formula exclusively based on vegetable lipids [54]. MFGM components such as phospholipids and mucin can promote the formation of binding groups on the surface of probiotics and corresponding uptake sites in intestinal cells [25]. The components of the MFGM can also display in vitro bactericidal activity against several foodborne pathogens, including *C. jejuni*, *Salmonella enteritidis*, and *Listeria monocytogenes* [55]. In vivo, rats supplemented with MFGM and then infected with *L. monocytogenes* were protected against pathogen colonization and translocation [55]. The secretion of microbiota regulators such as antibacterial proteins and immunostimulatory peptides is determined by the supramolecular structure of the MFGM [25]. In contrast, MFGM mucins inhibit enteropathogens that bind to host cell receptor glycans. The mucin sialic acid from *E. coli* and *Salmonella enterica* is an essential binding site for *MUC1*. The addition of MFGM can prevent *Helicobacter pylori* from ingesting the mucin gel of *MUC5AC* from the gastric mucosa [25]. Furthermore, the MFGM protein can regulate immune and inflammatory responses by increasing the abundance of butyrate-producing *Lachnospiraceae* [25]. Furthermore, the biological functions of lipid fractions in MFGM have been thoroughly investigated. This bioactivity encompasses the prevention of bacterial and viral infections, the reduction of cholesterol-induced steatosis, and the maintenance of gut health [56]. Additionally, both human and caprine MFGM have inhibitory effects on the adhesion of *Cronobacter sakazakii* and *S. enterica* [25]. Table 1 contains a brief description of the effects reported about the impact of different milk components on GM.

It should be considered that animal milks cannot always be consumed in its natural form, and to extend its shelf life, as well as to ensure its microbial safety, it often requires processing, including refrigeration, homogenization and heat treatment [58]. The most common heat treatments widely used in the dairy industry to achieve milk safety and preservation are pasteurization and ultra-high temperature (UHT) sterilization. Thermization and in-bottle sterilization are also performed on raw milk [59]. However, the heat treatments applied to milk not only ensure its microbiological safety but also modify the organoleptic properties and composition of the milk [59]. These effects on nutritional composition include the denaturation of some protein fractions such as whey proteins and enzymes, lactose degradation, and the inactivation of potentially functional components [60].

Both refrigeration, homogenization and thermal treatments can also affect milk microbiota [59]. Refrigeration time favors significant changes in the quantity and population composition of microorganisms in raw milk, increasing genera as *Xanthobacterium*, *Pseudomonas*, and *Lactococcus* [61]. Pasteurization and UHT destroy bacteria, toxin-producing and spore-forming organisms, and UHT destroys all vegetative microorganisms [59]. However, the thermal treatments also destroy lactic acid bacteria that are commonly in raw milk, such as lactobacilli or *Lachnospiraceae* [62].

Additionally, it should be noted that there are very important differences in the composition, structure and physicochemical properties of milk depending on the animal species from which it originates. Thus, in the case of protein content, although there are relevant differences in total protein content among CM and other milks such as goat, sheep of camel milk (CAM), milks from these latter species have a lower casein-to-whey-protein ratio as well as a relatively higher *β*-casein-to-*α*-casein ratio compared to CM [63]. The fat content varies significantly even within the same animal species depending on factors such as breed (3.3–5.4% in CM, 3–7.2% in goat milk (GOM) and 2–6% in CAM) [63]. GM is richer in SCFAs and medium-chain triacylglycerols than CM, whereas CAM contains a higher rate of long chain fatty acids and a lower rate of SCFAs than CM [64]. With respect to lactose content, CM contains higher content (4.4–5.6%) than GOM (3.2–5%) or CAM (3.5–5.1%) [63].

It should be considered that feeding using in livestock widely influences the composition of the milk not only in proximate composition but also in minor compounds such as terpenes, phytanic acid, pristanic acid, skatole, antioxidants of fatty acids [65].

### 3.1. Effects of Cow Milk on the Gut Microbiota

CM is one of the most consumed milk types worldwide. According to FAO data [66], worldwide CM consumption is 87 kg/person annually, representing an increase of 14% with respect to that in 2000. Particularly in Europe, North America, India, and Oceania, cow dairy products constitute a significant fraction of the daily diet [67].

The bioactive components of CM, especially oligosaccharides and whey proteins, such as lactoferrin, lysozyme and alpha-lactalbumin, have been shown to play crucial roles in shaping the GM from birth to adulthood [22]. Recent research on oligosaccharides has revealed substantial disparities in their composition profile and relative abundance among cows. Compared with mature milk, cow colostrum is considered a more desirable source of oligosaccharides because of its higher concentration of oligosaccharides and the simplicity with which it can be isolated and identified [68]. The bioactive oligosaccharides found in CM and colostrum closely resemble those found in human milk in terms of their chemical composition [68,69]. These oligosaccharides play crucial roles in several biological and physiological processes, such as prebiotic action and protection against different infections. Hence, they hold tremendous relevance [70,71]. Their primary role seems to be to act as competitive inhibitors for the binding sites on the surfaces of intestinal epithelial cells, thus providing protection against infections. Furthermore, evidence substantiates the notion that certain bioactive constituents function as promoters of the growth of the microflora in the colon [72]. Karav et al. [57] reported that CM oligosaccharides stimulate the growth of the *Bifidobacterium longum subspecies infantis* in newborns, which is comparable to the effect of human milk oligosaccharides. Glycans derived from CM and colostrum, which are complex and hybrid in nature, exhibit prebiotic activity and are specifically used by beneficial microbes. Cow milk oligosaccharides further enhance brain development and mitigate metabolic problems. Even though CM contains all the growth factors required by probiotics, it is not always necessarily available in acceptable forms or at optimal concentrations [73].

Various studies have investigated the effects of CM (anole or CM combined with other nutritional interventions) on the GM in both animal [2,74,75,76] and human [39,77] models (Table 2). In human models, the intake of CM was found to increase the counts of *Roseburia* and lactobacilli when the counts of *Prevotella* decreased [77]. Additionally, in babies that consumed CM, lower counts of *Erysipelotrichaceae* and *Bacteroidaceae* were detected [39]. Overall, these results could be considered beneficial since lactobacilli, *Roseburia* and *Prevotella* are beneficial species because they are SCFA producers [78]. SCFAs have been shown to increase postprandial concentrations of glucagon-like peptide 1 and peptide YY and thus increase satiety and reduce energy intake [10]. SCFAs also exhibit antimicrobial effects against *E. coli*, *L. monocytogenes* and *S. aureus* both in vitro and in vivo [54]. Immunologically, butyrate not only protects enterocytes from damage by enhancing intestinal barrier function but also promotes the production of Treg cells in the intestine. In addition, SCFAs can improve metabolic syndrome by promoting the secretion of peptide hormones, which indicates that advanced glycation end products can affect immune metabolism via the GM [6]. In addition, caprylic acid has been shown to have inhibitory effects on pathogens, both of which reduce bacterial growth [54]. In contrast, *Erysipelotrichaceae* can be considered a harmful family, since it is often related to inflammation-related disorders of the gastrointestinal tract and is enriched in colorectal cancer [79,80].

Among the relevant results published on the effects of cow milk intake on the GM in experimental animal models, milk constituents can promote the growth of beneficial probiotic bacteria, including lactobacilli and *Bifidobacterium* [7,22,24,75], and decreases in *Clostridium perfringens* and *Clostridium sensu stricto_1* [7]. Other works revealed that the protein content of CM was negatively correlated with the abundances of lactobacilli, *Bifidobacterium*, *Bacillus*, *Escherichia-Shigella*, *Akkermansia*, *Enterococcus*, and *Proteus* [24]. The fat content of cow milk was significantly positively correlated with *Clostridium_sensu_stricto_1* and was negatively correlated with *Veillonella*, *Peptoclostridium*, and *Akkermansia*. The total solid content was negatively correlated with lactobacilli, *Bifidobacterium*, *Akkermansia*, *Streptococcus*, *Enterococcus*, and *Bacteroides* and was positively correlated with *Peptoclostridium* [24]. *K*-casein-derived hydrolysates have been shown to stimulate the growth of *Bifidobacterium bifidum* in synthetic culture [7].

Furthermore, short peptides produced by the proteolytic digestion of beta-lactoglobulin from cow milk showed growth proliferation effects on *Bifidobacterium* and lactobacilli [7]. Furthermore, lactoferrin hydrolysates increased the growth of *Bifidobacterium adolescentis* B-1 in a dose-dependent manner [7]. In addition, previous in vitro studies have demonstrated the efficacy of glycomacropeptide (GMP), which is mostly found in dairy products and is released in whey by enzymatic digestion during cheese-making processes, inducing the growth of probiotics, such as several species from the *Bifidobacterium* genus [22]. GMPs also alter immune responses, inhibit digestive tract hormone activities, and regulate blood flow by exerting effects on hypertension and antithrombotic ability [81,82]. Additionally, GMP was also found to be capable of modulating elderly individuals, decreasing the abundance of *Clostridium* cluster IV and *Ruminococcus* 907. It also neutralizes the microbial toxins produced by *Escherichia coli* and *Vibrio cholerae* [83].

The results of the non-GM parameters investigated revealed a decrease in total cholesterol and HDL-c in CM-treated patients, as well as a decrease in proinflammatory factors such as TNFα, monocyte chemoattractant protein-1 and interleukins [84,85]. In other works, an increase in several immunoglobulin fractions was reported [2]. Additionally, a significant increase in SCFA production was detected after CM intake [75,76,85].

**Table 2 nutrients-16-03108-t002:** Effects of animal milk intake on human gut microbiota.

Model Work	Subjects	Dosage and Time of Exposition	Effects of Gut Microbiota	Other Health Effects	Reference
Human model	27 type 2 diabetic patients	10 g CAM power or CM power twice daily for 4 weeks	Significant increase in *Phascolarctobacterium* and decrease in unclassified *Micrococcaceae* for CM-treated patients. Significant increase in relative abundance of *Clostridium_sensu_stricto_1* in CAM-treated patients	Significant decrease in fasting blood glucose in patients intervened with CAM. Decrease in total cholesterol and HDL-c in both CAM and CM treated patients. Decrease in TNFα and MCP-1, especially in CM-treated patients. Decrease in resistin and lipocalin-2 levels in CAM-treated patients	[84]
Human model	90 babies	Breast milk, GOM or CM-based formula for 4 months	α-diversity were less diverse in breast milk-fed children than in formula-fed babies. *Erysipelotrichaceae* were less abundant in breast milk-fed infant microbiotas, whereas *Bacteroidaceae* were more abundant	Not provided	[39]
Human model	96 overweight or obese people	500 kcal daily restriction diet either high (1500 mg Ca/day) or low (≤600 mg Ca/day) in dairy products for 24 weeks	*Veillonella* genus was significantly decreased in low dairy group	Not provided	[10]
Human model	64 male subjects	500 mL of low glycinin soymilk, conventional soymilk, or CM daily for 3 months	Decrease in Proteobacteria phylum in all groups. People who consumed CM increased counts of *Roseburia* and decreased the counts of *Prevotella*. *Lactobacilli* increased in subjects who consumed CM, while they decreased in individuals whose consumed conventional or low-glycinin milk	Not provided	[77]
Rats model	48 Sprague–Dawley rats (12 per group)	Restricted caloric diet (5 g/100 g) body weight. Control group received a standard diet, whereas the 3 other groups received 40% of diet by kay, CM or CAM for 28 days	Rats feed with cow or yak milk decreased their GM diversity, whereas rats feed in camel milk increases diversity. Patterns of microbial changes on day 28 was very similar across all three milk groups, featured with less *Ruminococcus*, *Prevotella*, *Barnesiella intestiniformis*, and more *Blautia*, *Bacteroides*, *Parabacteroides* and *Clostridium*. From the GM point of view, yak and CAM are healthier to consume than CM	Interferon-γ levels were significantly higher in rats feed with CAM. Rats fed with CM increased levels of IgA, IgG and IgM	[2]
Mice model	70 C57BL/6J mice	10 mL/kg body weight daily for 4 consecutive weeks, intragastrically. Seven groups were made: distilled water; whole goat milk; milk fat; casein; milk whey; whey protein	Mice fed with whole goat milk and casein fraction showed higher gut microbiota richness that distilled water-treated mice. Diversity was lower in the whole goat milk and fat milk groups. Whole goat milk increased the relative abundance of *lactobacilli*	It was found that treatment with certain milk fractions reduced significantly the relative abundance of genes involved in endocrine, cancerous and infectious diseases	[86]
Rats model	50 Sprague–Dawley rats (10 per group)	*Ad libitum* access to water, casein in water, CM, soy beverage or almond beverage	Increase in Actinobacteria phyla (*Coriobacteriaceae* and *Bifidobacteriaceae*) and decrease in Bacteroidales (*Porytomonadaceae* and *Bacteroidales* S24-7 group) and Firmicutes (*Peptostreptococcaceae*) phyla in milk-treated animals with respect to water or vegetal beverages-added animals. *Lachnospiracease* was higher counts in milk-added animals than in vegetal beverages-added animals	Increase in body weight of soy-added animals than in milk or casein-added animals, and these were higher than in almond or water-added animals	[74]
Mice model	32 C57BL/6J mice	10 mL/kg body weight daily for 4 consecutive weeks for 21 days, and the same with 2.5% dextran sodium sulfate, control group and a group without CAM and with 2.5% dextran sodium sulfate	Increasing in GM diversity, and SCFAs production, increase in beneficial bacteria such as *Lachnospiraceae* and *Muribaculaceae*, and decrease in harmful bacteria as *Bacteroides*, and *Escherichia*-*Shigella*	Reduction in IL-1B, IL-6 and TNFα in mice administered with CAM. Inhibition of apoptosis of intestinal cells and promotion of the expression of claudin-1, occluding and zonula occludens proteins	[85]
Mice model	24 C57BL/6J mice	CAM at 3 g/kg body weight for 8 weeks	CAM increases cecal microbial *α*-diversity compared to alcohol-treated mice. Increasing of *Muribaculaceae*, *Lachnospiraceae*, *Blautia* and *Mucispirillum*	CAM prevented alcohol-induced colonic disfunction and lipid accumulation, regulated oxidative stress and inflammatory cytokine production	[87]
Mice model	6 BALB/c mice	Fresh GOM at 5 mL/day/mice for 4 weeks	Improved GM richness. Increase in Firmicutes/Bacteroidetes ratio. Increase if *norank_f_Bacteroidales_S24-7* group	Not provided	[17]
Rats model	60 Sprague–Dawley rats	Rats with dysbiosis induced by amoxicillin (50 mg/kg) were feed with whole CM or GOM for 14 days	Goat milk increased *Bifidobacterium*, lactobacilli and decreased *Clostridium perfringens*. CM increased lactobacilli and decreased *C. perfringens*	SCFAs increasing in rats fed with both goat and CM is a different way, but in higher proportions in the cased of goat milk-fed rats	[75]
Mice model	64 C57BL/6J mice	45 mL raw CAM daily for 28 days	Dromedary CAM propagated the beneficial bacteria (*Allobacterium* and *Akkermansia*) and reduced harmful bacteria such as Proteobacteria, *Erysipelotrichaceae*, and *Desulfovibrionaceae*	Weight gain in milk consuming mice	[79]
Mice model	24 Balb-c mice	120 g lyophilized milk contained of A1A2 or A2A2 CM or control diet daily for 4 weeks	*Deferribacteriaceae* and *Desulfovibrionaceae* as the most discriminant families for the A2A2 group, while *Ruminococcaceae* were associated with the A1A2 group	Increase in SCFAs, especially for isobutyrate	[76]
Mice model	12 C57BL/6J mice	10 mL of CAM or distilled water/kg body weight intragastrical once a day for 4 weeks	α-diversity increased in animals after fed CAM. Mice fed with CAM showed higher abundance in *Allobaculum*, *Akkermansia* and *Bifidobacterium* genera	Not provided	[88]
Mice model	24 C57BL/6J mice	10 mL of different raw or heat-treated CAM/kg body weight intraperitoneal once a day for 4 weeks	α-diversity in mice GM decreased proportionally to the heat treatment applied to milk. Beneficial genus as *Bifidobacterum* were lower in mice fed CAM with more severe heat treatments	Increase in SCFAs	[85]
In vitro	Fecal samples from 10 healthy infant donors were used for fermentations	Human breast milk, infant formula milk, CM, CAM, GOM and mare milk	Compared to initial values, the richness of microbiota of all kinds of milks except infant formula increased their richness. Proteobacteria counts decreased in all milks. *Akkermansia* decreased in all milks except mare milk. Mare milk also increased counts of *Bifidobacteriaceae*, *Lachnospiraceae*, and *Lactobacillae* more than other milks.	CAM and infant formula produced highest gas pressure than mare milk, human milk, and CM	[3]

CAM: camel milk; CM: cow milk; GM: gut microbiota; GOM: goat milk; HDL-c: high-density lipoprotein cholesterol; IL: interleukin; MCP-1: monocyte chemoattractant protein-1; SCFAs: short-chain fatty acids; TNFα: tumor necrosis factor alpha.

### 3.2. Effects of Goat Milk and Mare Milk on the Gut Microbiota

Mare milk is more like human milk in terms of microbial community functions [24]. Recently, researchers have focused on mare milk because of its high nutritional value, which is an optimal substitute for human milk and CM for minimizing allergies and hyperlipidemia-related complexities [17], and because of its high nutritional value, mare milk has been used as a substitute for CM for children who are allergic [89]. In terms of the GM, an in vitro study revealed that the feces of infants fed mare milk had greater abundances of the lactobacilli, *Bifidobacterium*, and *Akkermansia* genera than did the feces of infants whose milk fermented other milks. Indeed, the abundance of *Bifidobacterium* significantly increased in all groups, especially in the groups fermented with CM, mare milk, and infant formula [24]. Furthermore, lactobacilli also increased in all groups but mostly in the group fermented with mare milk [24]. 

The worldwide production of goat milk increased by 80% from 1991 to 2013 [89]. In terms of composition, goats contain similar natural oligosaccharides in CM although at lower concentrations and with less diversity than human milk does [67]. GOM also has 4 and 10 times more oligosaccharides than cow and sheep milk, respectively, and its chemical composition is closer to that of human milk [90]. The functions of free oligosaccharides in GOM are more like those in human breast milk than to those in CM [15]. Fermentable oligosaccharides cannot be digested by human enzymes in the small intestine but are extensively fermented into SCFAs in the large intestine [15]. SCFAs serve as energy sources for host cells and the intestinal microbiota, and they reduce systemic inflammation and improve lipid and glucose metabolism by enhancing intestinal barrier function [15]. In addition, SCFAs can increase mineral absorption and prevent the development of large intestinal diseases, such as ulcerative colitis and colorectal cancer [15]. GOM proteins were reported to exhibit antimicrobial and anticancer properties and may have a positive effect on the bioavailability of minerals, particularly iron and calcium [91].

In the GM, the effects of the intake of GOM or its components have been investigated via in vitro assays [3], mouse models [86,92], rat models [51], and human models [39]. The results revealed that GOM fat treatment induced a greater proportion of *Helicobacter*, which is associated with gastrointestinal diseases, such as gastric inflammation, peptic ulcers and even gastric cancer [86].

Several studies have shown that GOM can positively modulate the intestinal microbiota and thus provide beneficial effects for the host. A paper analyzing the infant fecal microbiota revealed that whole GOM was the only group in which the relative abundance of lactobacilli increased significantly after treatment [86] but decreased or was unaltered when the GOM fractions were administered separately. Lactobacilli has been reported to increase the production of anti-inflammatory metabolites in host intestinal epithelial cells, which contributes to the first line of defense against enterovirulent bacteria [86]. In addition, the *Lactobacillus* and *Lactococcus* genera are negatively associated with potential pathogens as *Helicobacter* and *Acinetobacter* [86]. The ingestion of the whey fraction has also been shown to increase the abundance of *Blautia*, which produces ethanol, acetate, succinate, and lactate as end products of glucose fermentation and is associated with multiple biological activities, such as anti-inflammatory activity, energy homeostasis, and satiety [86]. In addition, the use of whole GOM or specific GOM fractions may provide prebiotic benefits for the maturing gut development of formula-fed infants [90].

In addition, *Helicobacter* infection also increases the incidence of metabolic diseases, such as metabolic syndrome, diabetes, and nonalcoholic fatty liver disease [86]. However, goat serum has natural antimicrobial activity and antitumor potential due to the presence of immunoglobulin, lactoferrin, lysozyme and partially digestive peptides [86]. In addition, GOM has special characteristics that distinguish it from CM, such as better digestibility, greater alkalinity and greater buffering capacity [91]. In terms of nutrients, GOM has greater bioavailability of minerals and more balanced protein and fat profiles than CM does [73]. In addition, the composition of GOM is closer to that of breast milk than to that of other milk sources [92].

Bifidobacteria were the most abundant microbes in the feces of 2-month-old infants fed human milk, whole GOM, or whey-based CM [90]. Compared with those of infants fed human milk or whey-based cow milk formula, the microbiota DNA sequences of infants fed human milk or whole GOM formula were more similar [90].

Rats that consume GOM also have increased numbers of *Lachnospiraceae* [75], which are important SCFA-producing bacteria that act as tumor suppressors and might elicit the activation of the immune system [93]. The number of bifidobacteria and lactobacilli was significantly greater in the group that consumed GOM than in the group that did not receive antibiotics [75]. Additionally, in the same work, the abundance of *C. perfringens* decreased in the cecum [75]. Furthermore, the GM was restored in animals that consumed CM and GOM within two weeks of disruption by antibiotic administration [75]. In fact, GOM effectively increased lactobacilli and *Bifidobacterium* while decreasing *C. perfringens* in rats with amoxicillin-induced intestinal dysbiosis [75].

Furthermore, GM dysbiosis affects host immunity and metabolism through SCFAs. In addition, 60% more total SCFAs are increased in animals that consume GOM than in control animals [75].

### 3.3. Camel Milk

For centuries, ancient peoples used camels, including dromedary and Bactrian camels, for milking and transportation, especially in Africa, the Middle East, Asia, and India [89]. Currently, the global production of CAM is estimated to be 5.4 million tons [89].

CAM has a unique composition that differs from the milk of other ruminants. It contains higher levels of immunoglobulin, lactoferrin, and calcium and lower levels of fat. Moreover, CAM contains a variety of secreted antibodies, such as IgM and IgA [94]. Like human milk, CAM lacks *β*-lactoglobulins, and α-lactalbumin is the major whey protein in CAM [38]. Among all mammalian milk species, CAM fat globules are the smallest, and they do not physically gather due to the lack of agglutinin substrate [38]. The lactose content of CAM is comparable to that of CM and contains a higher concentration of L-lactate than other milk, such as CM; this may contribute to its lower lactose intolerance than that of CM [38,87]. Additionally, CAM contains several nanoantibodies with marked antibacterial and antiviral activities. It also contains various bioactive proteins with immunomodulatory properties, including lysozymes, lactoperoxidase, and *N*-acetyl glucosinidase [87]. Moreover, CAM is rich in lactoferrin, which is a protein with marked antioxidant and anti-inflammatory properties [87]. CAM can prevent body weight loss, and colon milk can attenuate colon tissue damage [94], reduce the overexpression of inflammatory factors, inhibit the apoptosis of intestinal epithelial cells, and promote the expression of the claudin-1, occludin and zonula occludens-1 proteins.

In terms of its effects on the GM, CAM has been investigated in both in vitro [3], animal [79,85,87,88,94] and human models [84] (Table 2). It has been reported that CAM oligosaccharides are essential for improving the proliferation of intestinal bifidobacteria in addition to effectively inhibiting the adhesion of pathogenic microorganisms to the colonic mucosa [38]. Thus, beneficial effects on the GM composition were expected to be obtained after CAM intake. CAM peptide intervention markedly reversed gut microbiota dysbiosis in type 2 diabetic mice by reducing the relative abundance of Proteobacteria, *Allobacterium*, *Clostridium*, and *Shigella* and the Firmicutes/Bacteroidetes ratio while increasing the relative abundance of SCFA producers such as Bacteroidetes and *Blautia* [93].

CAM also increased GM diversity and the abundance of beneficial bacteria such as *Lachnospiraceae* and *Muribaculaceae* [94], *Allobacterium*, *Akkermansia* [79,88,95], and *Bifidobacterium* [88]. *Lachnospiraceae* are intestinal bacteria that produce SCFAs such as butyric acid [94]. *Muribaculaceae* are positively related to the barrier function of the intestinal mucus layer, play a role in the degradation of complex carbohydrates and produce both acetic and propionic acid [94]. Propionic acid has been reported to protect the intestinal mucosa and suppress inflammatory cytokine production [94]. *Allobaculum* may have an intimate relationship with obesity and could be considered an indicator bacterial genus for obesity [88]. *Allobaculicum* was also negatively correlated with adiposity [96]. *Allobaculicum* has several health effects, such as SCFAs production and obesity control [79]. In fact, a study confirmed the negative correlation between *Allobacterium* and SCFAs as well as HDL-c [93]. Once the permeability of the epithelium increases, pathogenic bacteria and antigens invade cells, which activates the host immune system [94]. The effect of CAM on the growth of *Anaerostipes* and *Clostridiales*, in addition to its relationship with increased production of SCFAs in the gut and the immune system response, is now under consideration.

In contrast, CAM intake has been reported to reduce the number of potentially harmful bacteria, such as Bacteroides, *Escherichia*-*Shigella* [94], *Erysipelotrichaceae*, Proteobacteria, and *Desulfovibrionaceae* [78]. Additionally, CAM can inhibit bacteria such as *Bacillus*, *Candida*, *Diplococcus*, *Klebsiella*, *Listeria*, *Pseudomonas*, *Salmonella*, *Staphylococcus* and *Streptococcus* [78,88,95]. Other genera, including *Turicibacter*, *Pseudomonas*, *Lachnoclostridium*, and *Alistipes*, are also reduced following the intragastric administration of CAM, indicating that CAM could inhibit the growth of these bacteria [88].

Additionally, CAM intake has other beneficial effects on human health, such as significantly decreasing fasting blood glucose and resistin and lipocalin-2 levels [69], increasing interferon-γ levels [2], increasing SCFA production [94], reducing TNFα [94], resistin and lipocalin-2 levels [84], preventing alcohol-induced colonic disfunction and lipid accumulation, and regulating oxidative stress and inflammatory cytokine production [94].

## 4. Vegetable Beverages and Effects on the Human Gut Microbiota

Vegetable milks, correctly called vegetable drinks, have been identified as close replicas of dairy milk in terms of physical and organoleptic attributes. Vegetable drinks can contain a wide variety of water extracts of disintegrated or dissolved vegetable materials, such as pseudocereals, oil seeds, tubers, cereals and legumes [1]. In addition, the functional contributions of protein and calories are claimed to be like those of milk of animal origin [1].

Plant-based beverages differ in their composition and nutritional value from animal-origin milks in terms of its content of proteins, lipids and carbohydrates, and glycemic index [12]. The nutritional value of plant-based beverages varies depending on the raw material from which they are produced and the production technology employed [12]. Nevertheless, in most cases, plant-based beverages are high in carbohydrates and low in protein, containing up to 30 times less protein that CM. Additionally, the protein of plant-based beverages contains lower amount limiting amino-acids (lysine in cereals, methionine in legumes) and poor digestibility than animal-origin milks [12,97]. Plant-based milk substitutes can be also fermented to obtain diary-free yogurt-type products while rendering the raw material into a more palatable form [98].

In most cases, plant-based beverages are low in fat unless supplemented with vegetable oils. Compared to milk, plant-based beverages have a lower content of saturated fatty acids (SFAs), except for coconut beverages, which are SFA-rich. Plant-based beverages are dominated by unsaturated fatty acids, mainly in the form of oleic, linolenic, and linoleic acids [12,97]. Animal-origin milks are natural sources of calcium, and plant-based beverages usually content lower amount of this nutrient if not fortified during production. Animal-origin milks contain naturally-occurring vitamin A and trace amounts of vitamins D, E, K, C and B. It is also a source of phosphorus, potassium, zinc and easily digestible magnesium as well as small amounts of sodium and iron [99].

Regarding the special benefits of plant-based beverages, they contain bioactive ingredients with health-promoting effects, such as ß-glucans, phytosterols and polyphenols [12,100]. Plant-based beverages does not contain lactose or cholesterol [12]. Finally, they are rich in antioxidants, which helps to decrease oxidative stress in the body [12].

The protein digestibility of vegetable beverages is usually greater than that of vegetable products [74]. However, although vegetable beverages can contain fewer nutrients that dairy milks, their health benefits based on their phytonutrient composition have remarkably promoted their appraisal and recognition as functional foods [1]. In contrast to animal milk, in which lactose is the predominant carbohydrate, the predominant carbohydrates in vegetable drinks are non-starch polysaccharides and sugars [63]. Therefore, plant-based beverages may represent an ideal vehicle for supplying probiotics to consumers with milk protein allergies or severe lactose intolerance [74]. The high activity of lactic acid bacteria during the lactic acid fermentation of vegetable beverages causes similar changes in the product composition to that observed in lactic acid bacteria-fermented milk, including the formation of organic acids, acidification of the environment, decomposition of some carbohydrates, and digestion of proteins and lipids. Thus, fermented vegetal beverages can be an alternative to fermented milk to meet the growing demand for this type of product among consumers [101].

The substitution of animal-derived milk with vegetable drinks was reported to provide some health benefits, such as reducing the risk of overweight and obesity, type 2 diabetes, cardiovascular diseases, and certain types of cancer [102]. The dietary intake of soy and fiber is associated with decreased blood cholesterol levels 104]. On the other hand, it was reported that there could be some concerns regarding plant-based formulas and their possible effects on sexual development and reproduction, neurobehavioral development, immune function, and thyroid function [11].

Owing to the global trend in several parts of the world of an increase in both vegetarian and vegan people [102], the number of scientific articles investigating the effects of these beverages on human health has increased dramatically in recent years. In terms of their effect on GM, much attention has been given to soy beverages [11,74,77,103,104,105,106,107], whereas others, such as almond drinks, have received less attention [74] (Table 3).

In global terms, the results obtained in the GM after the administration of vegetable beverages can be considered beneficial. Some beneficial species, such as *Bifidobacterium*, are increased in various works [105,106,107,108]. The same result was obtained for lactobacilli [11,74,103,106,108] or *Blautia* [103]. A previous study reported that compared with CM protein intake, soy protein administration resulted in beneficial changes in the GM [104].

However, the results also revealed some results cannot be considered as beneficial to the host. In this sense, an increase in *Fusobacterium* and *S. enterica* was detected in piglets fed a plant-based formula [11]. *Fusobacterium*, especially *Fusobacterium nucleatum*, is a bacterial genus associated with the development of colon cancer [109], whereas *S. enterica* is a well-known foodborne pathogen [110,111]. An increase in the Firmicutes-to-Bacteroidetes ratio was also observed [103], which is commonly related to obesity [87]. In some cases, a decrease in both lactobacilli and *Bifidobacterium* [77], as well as an increase in Proteobacteria, a phylum that includes several foodborne pathogen species and has been associated with obesity and dysbiosis in human populations, has been reported [74]. The increase in Proteobacteria, especially *Enterobacteriaceae*, is not consistent since in other works, a decrease in the GM was detected after soymilk intake [107]. Finally, in other works, no significant changes in the GM were detected after soymilk intake in a mouse model [112].

Comparing individually the results obtained for specific vegetable beverages, it was found that almond milk consumption achieved in a rat model a higher relative abundance in Bacteroidetes (*Bacteroidaceae*) and Firmicutes populations (*Lactobacillaceae*, *Clostridiaceae* and *Peptostreptococcaceae*) than soymilk or CM consumption [74]. Bacteroidetes members have gene-encoded carbohydrate active enzymes than can switch readily between different energy sources in the gut, depending on availability, whereas some genus of Firmicutes were described as SFCAs producers [74]. Interestingly, the results obtained by the same work regarding bone density prevention showed similar results in CM, soy beverages and almond beverages [74]. This result is important because it suggests that vegetable drinks, despite the lower content of calcium than animal-origin milks, can play a positive role in maintaining bone density.

**Table 3 nutrients-16-03108-t003:** Effects of vegetable beverages intake on gut microbiota.

Model Work	Subjects	Dosage and Time of Exposition	Effects of Gut Microbiota	Other Health Effects	Reference
Rat model	50 Sprague–Dawley rats (10 per group)	*Ad libitum* access to water, casein in water, bovine milk, soy beverage or almond beverage	Proteobacteria were higher in soy beverage-added animals. Increase in *Acidobacteria* in almond group. *Lactobacillaceae* were higher in soy and almond-treated groups than in the water or milk-added groups. Proteobacteria family member *Enterobacteriaceae* was higher in soy-supplemented group than in almond or milk-added groups	Bone density results from our study suggests milk and soy supplementations are equally beneficial for (bone) health	[74]
Piglet model	18 pigs (9 per group)	1.047 MJ/kg/day of dairy-based formula or plant-based formula for 11 days	No differences were found for *β*-diversity between dairy- or plant-based fed piglets. *Lactobacillus delbrueki*, *Lactobacillus crispatus*, *Fusobacterium* and *Salmonella enterica* were higher in the GM of piglets fed with plant-based formula	Both pro- and anti-inflammatory cytokines, minerals, vitamins and hormones measured in plasma of piglets showed no significant differences between dairy- and plant-based fed piglets	[11]
Rat model	35 Sprague–Dawley rats	*Ad libitum* access to control diet, soymilk diet, high fiber diet or high cholesterol diet for 6 weeks	Soymilk increased the Firmicutes-to-Bacteroidetes ratio due to an increase in lactobacilli counts. Increased genus *Coprococcus* and *Blautia* and decreased *Barneisella* spp.	Soy diet improved serum HDL-c, and expression of ZO-1 and Occludin genes and inflammation-related proteins	[103]
Rat model	60 Wistar rats	3 mL soy product/kg of body weight/day for 30 days	Increase in total anaerobes, *Bifidobacterium*, *Clostridium*, *Enteroccocus* and lactobacilli	Not provided	[108]
Rat model	40 Wistar rats	2 mL soymilk/animal for 4 weeks	Not significant changes were observed	Not provided	[112]
Human model	64 male subjects	500 mL of low glycinin soymilk, conventional soymilk, or bovine milk daily for 3 months	Firmicutes-to-Bacteroidetes ratio decreased in low-glycinin soymilk, conventional soymilk treated subjects. Decrease in Proteobacteria phylum in all groups. Lactobacilli decreased in individuals whose consumed conventional or low-glycinin milk	Not provided	[77]
Human model	12 infants	Soy formula (exclusive feeding) for 1 month	This feeding decreased the intestinal bifidobacterial population	Not provided	[105]
Human model	6 male subjects and 4 female subjects	100 g/day of nonfermented soymilk for 2 weeks	Increased of lactobacilli and *Bifidobacterium*, and decreased counts in *Clostridium* after soymilk intake	Not provided	[106]
Human model	4 male subjects and 4 female subjects	100 g/day of soymilk for 28 days	Increase in *Bifidobacterium* and decrease in *Enterobacteriaceae* on GM	Not provided	[107]

Regarding soy beverages, most of the results obtained about its effects on GM were beneficial, including increases in lactobacilli [11,74,106] and *Bifidobacterium* [107,108,112], or decreases in Proteobacteria [77,107]. Most of results were beneficial, with the few exceptions of increases in *Salmonella enterica* or *Fusobacterium* [11]. In both cases, the authors of the work claim that their increases have been in low proportions and that they still represented a minor percentage of the bacterial groups present in the GM [11].

## 5. Conclusions

The consumption of vegetable milk substitutes is increasing worldwide, especially in some geographical areas, such as Europe.

However, in addition to the macronutrient and calcium contents, which plant-based beverages mimic, milk contains numerous minor compounds, a unique structural composition of fats and a mixture of protein fractions that can affect the human GM. There are important differences in the protein content and composition, lipid composition, absence of lactose and cholesterol, and different amounts of minor components. It should be also considered that the composition of animal milk varies greatly depending on factors such as the species and breed of animal from which it comes, the animal’s feed and the industrial treatment it undergoes. Vegetable beverages, on the other hand, have a more stable composition, since their initial composition and technological treatment is usually stable.

With respect to the effects of both types of beverages on the intestinal microbiota, the first conclusion that can be drawn from this work is that there is little information on the subject. Few studies have tested the effects of animal milk or plant-based beverages per se (without added prebiotics or probiotics) on the GM, especially in humans. Thus, knowledge about the effects of milk and plant-based beverages on the GM is still incomplete, and much more work is needed before an adequate consensus can be established.

Based on the results shown so far, it appears that the consumption of milk of animal origin exerts beneficial effects on human GM. In contrast, vegetable beverages also show mostly positive results, but in some cases, they also favor the growth of potentially negative bacterial genera. Therefore, at least from the point of view of their effects on the GM, it cannot be said that plant-based milk replacers are a perfect substitute for milk, the latter being generally more beneficial for the composition of the human GM.

All this needs to be considered with the due limitations, since these are in vitro studies in animals or with few human subjects. The effects of milk and plant-based drinks on the GM may vary from person to person, and therefore, before recommending the population to opt for milk of animal origin or its vegetable substitutes, it is very important to adopt an individual nutritional approach that considers the needs and predispositions of each consumer.

## Figures and Tables

**Figure 1 nutrients-16-03108-f001:**
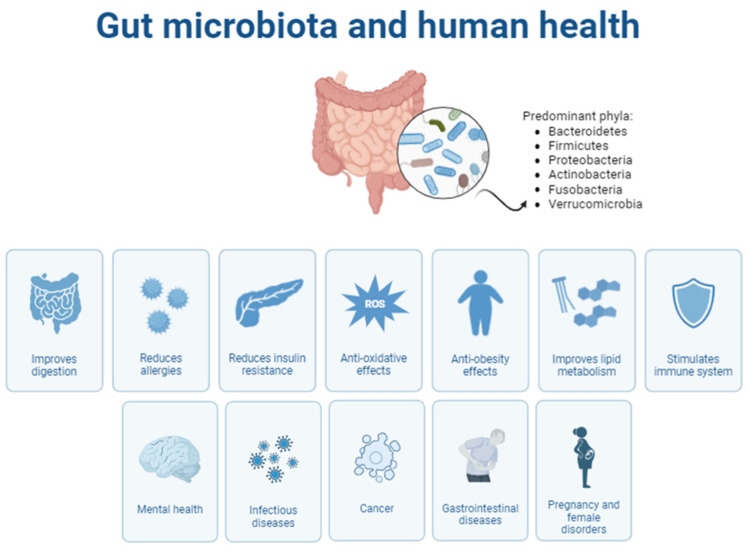
Gut microbiota and its influence in different aspects of human health.

**Table 1 nutrients-16-03108-t001:** Effects of animal milk components on human gut microbiota.

Milk Component	Effects on Gut Microbiota	References
Lactose	Increase lactobacilli and *Bifidobacterium* growth	[7,24]
Milk oligosaccharides	Favors growth of beneficial bacteria as lactobacilli and *Bifidobacterium*; Inhibiting bacterial adhesion of pathogens to enterocytes	[22,25,57]
α-lactalbumin	Promoting growth of beneficial bacteria and exerts antimicrobial activity against some pathogens	[22]
Lactoferrin	Prebiotic effect and inhibition of pathogens	[22,41,48,49]
Lysozyme	Increase resistance to intestinal colonization by some pathogens	[44]
Milk fat globule membrane	Promote the formation of binding groups on the surface of probiotics; In vitro bactericidal effects against some pathogens	[25,46,54,55]

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
