# Peer review of "Substitutive Effects of Milk vs. Vegetable Milk on the Human Gut Microbiota and Implications for Human Health"

_nutrients, 2024, doi:10.3390/nu16183108_

Round 1

Reviewer 1 Report

Comments and Suggestions for Authors

In recent year, there is a trend to replace animal milk with plant-based dairy products. In this context, the authors compared the the effects of animal milk and vegetable beverages on the microbiota composition in different in vitro, animals and human assays. The topic of this review is interesting. However, some modifications are required:

Abstract: The authors have introduced too much background information. However, the detailed impacts of different milks on the human gut microbiota are not clearly introduced.

Keywords: Replace the last two keywords with others.

Introduction: The aim and content of this review is not introduced in the last section of introduction.

The human gut microbiota: In this section, it is recommended to add a figure to illustrate the composition of gut microbiota and their relationship with human health.

Effect of animal-based milk on the gut microbiota: In this section, a table is needed to summarize the impact of different milk components on gut microbiota. Also in this section, the components and their contents of cow milk, goat milk and camel milk should be compared. Meanwhile, their impacts on the gut microbiota should also be compared and discussed.

Vegetable beverages and effects on the human gut microbiota: This section is too simple and should be carefully improved, especially the differences in the compositions between animal-based milk and vegetable beverages. Also the impacts of animal-based milk and vegetable beverages on human gut microbiota should be carefully compared and discussed. Please note the composition of animal-based milk and vegetable beverages are very different. The impact of each component in vegetable beverages on gut microbiota should be introduced.

Comments on the Quality of English Language

Minor editing of English language required.

Author Response

Reviewer 1:

Comments 1: In recent year, there is a trend to replace animal milk with plant-based dairy products. In this context, the authors compared the the effects of animal milk and vegetable beverages on the microbiota composition in different in vitro, animals and human assays. The topic of this review is interesting. However, some modifications are required

Response 1: The authors sincerely thank the constructive words from the Reviewer. In fact, we think that the Review contains interesting information, but it is indiscousable that after the Review process the manuscript was strongly improved.

Comments 2: Abstract: The authors have introduced too much background information. However, the detailed impacts of different milks on the human gut microbiota are not clearly introduced.

Response 2: That you for your comments. According to your suggestion, the introducing information in the abstract was shortened. In concrete, the following paragraph were deleted: “This increase is due to several factors, such as the increase in the population allergic to milk proteins or self-perceived lactose intolerance. In addition, some ethical aspects have also been promoted, with an increase in vegetarianism and veganism, whereby some people perceive the dairy exploitation of animals as cruel. There are also people who substitute animal milk in the context of reduced saturated fat intake.”

And according to your suggestion, the following paragraph was added: “Thus, it was found a increase of GM richness and diversity, increase in the production of short chain fatty acids, and beneficial microbes such as Bifidobacterium, lactobacilli, Akkermansia, Lachnospiraceae or Blautia. In other cases, significant decrease in potential harmful bacteria as Proteobacteria, Erysipelotrichaceae, Desulfovibrionaceae or Clostridium perfingens were found after animal-origin milk intake.”

Comments 3: Keywords: Replace the last two keywords with others.

Response 3: According to the suggestion from the reviewer, in the revised version of the manuscript the keywords “alpha-diversity and Short chain fatty acid” were changed to “milk oligosaccharides, milk fat globule membrane”.

Comment 4: Introduction: The aim and content of this review is not introduced in the last section of introduction.

Response 4: In fact, the aim of the Review was previously placed after section 2 in the original version of the manuscript. According to your comments, it was changed to the end of the Introduction section the following paragraphs:

“In recent years, many papers evaluating the effects of the consumption of milk with probiotics or prebiotics on the human GM have been published [18-22]. Similarly, vegetable beverages supplemented with probiotics [23-25] or components from milk, such as dairy proteins [12,26], dairy fats [27-29], or vegetable proteins from vegetable beverages [30-32], were also recently published.

However, despite its common consumption, the number of articles that have investigated the effects of animal milk and its natural vegetable substitutes on the GM is relatively low. Thus, the aim of this narrative literature search was conducted up to June 2024 for all the available literature in the Web of Science and Scopus. A combination of the following search terms was applied: “milk”; “milk substitute”; “vegetal beverages”; “plant-based milk” as the topic; and “gut microbiota” in the title. A total of 64 articles were ultimately selected and included in the review, as were the other 48 articles that were included to contextualize the article.”

Comment 5: “The human gut microbiota: In this section, it is recommended to add a figure to illustrate the composition of gut microbiota and their relationship with human health.”

Response 5: According to the suggestions from the Reviewer, a Figure (Figure 1) was included in the revised version of the manuscript.

Comment 6: Effect of animal-based milk on the gut microbiota: In this section, a table is needed to summarize the impact of different milk components on gut microbiota. Also in this section, the components and their contents of cow milk, goat milk and camel milk should be compared. Meanwhile, their impacts on the gut microbiota should also be compared and discussed.

Response 6: According to the suggestion from the reviewer, a new table (Table 1) was included in the revised version of the manuscript:

Table 1 comprised a brief description of the effects reported about the impact of different milk components on GM.

Table 1. Effects of animal milk components on human gut microbiota.

Milk component

Effects on gut microbiota

References

Lactose

Increase lactobacilli and Bifidobacterium growth

[7,27]

Milk oligosaccharides

Favors growth of beneficial bacteria as lactobacilli and Bifidobacterium; Inhibiting bacterial adhesion of pathogens to enterocytes

[12,28,59]

α-lactalbumin

Promoting growth of beneficial bacteria and exerts antimicrobial activity against some pathogens

[12]

Lactoferrin

Prebiotic effect and inhibition of pathogens

[12,35,43,44]

Lysozyme

Increase resistance to intestinal colonization by some pathogens

[39]

Milk fat globule membrane

Promote the formation of binding groups on the surface of probiotics; In vitro bactericidal effects against some pathogens

[28,41,49,50]

Regarding cow, goat and camel milk composition, the following information was added to the revised version of the manuscript:

“Additionally, it should be noted that there are very important differences in the composition, structure and physicochemical properties of milk depending on the animal species from which it originates. Thus, in the case of protein content, although there are relevant differences in total protein content among CM and other milks such as goat, sheep of camel milk (CAM), milks from these latter species have a lower casein-to-whey-protein ratio as well as a relatively higher β-casein ratio-to-α-casein ratio compared to CM [63]. Fat content varies significantly even within the same animal species depending on factors such as breed (3.3-5.4% in CM, 3-7.2% in goat milk (GOM) and 2-6% in CAM) [63]. GM is richer in SCFAs and medium chain triacylglycerols than CM, whereas CAM contains a higher rate of long chain fatty acids and a lower rate of SCFAs than CM [64]. With respect to lactose content, CM contains higher content (4.4-5.6%) than GOM (3.2-5%) or CAM (3.5-5.1%) [63].

It should be considered that feeding using in livestock widely influences the composition of the milk, not only in proximate composition but also in minor compounds such as terpenes, phytanic acid, pristanic acid, skatole, antioxidants of fatty acids [65].”

Comment 7: Vegetable beverages and effects on the human gut microbiota: This section is too simple and should be carefully improved, especially the differences in the compositions between animal-based milk and vegetable beverages.

Response 7: Thank you very much for your comment. In the revised form of the manuscript, the following information regarding the composition of plant-based beverages and its differences with animal-origin milks was added:

“Plant-based beverages differ in their composition and nutritional value from animal-origin milks, in terms of its content of proteins, lipids and carbohydrates, and glycemic index [12]. Nutritional value of plant-based beverages varies depending on the raw material from which they are produced, and the production technology employed [12]. Nevertheless, in most cases, plant-based beverages are high in carbohydrates and low in protein, containing up to 30 times less protein that CM. Additionally, protein of plant-based beverages contains lower amount limiting amino-acids (lysine in cereals, methionine in legumes) and poor digestibility than animal-origin milks [12, 98]. Plant-based milk substitutes can be also fermented to obtain diary-free yogurt-type products while rendering the raw material into a more palatable form [99].

In most cases, plant-based beverages are low in fat unless supplemented with vegetable oils. Compared to milk, plant-based beverages have a lower content of saturated fatty acids (SFA), except for coconut beverage, which is SFA-rich. Plant-based beverages are dominated by unsaturated fatty acids, mainly in the form of oleic, linolenic, and linoleic acids [12,98]. Animal-origin milks are natural sources of calcium, and plant-based beverages usually content lower amount of this nutrient if not fortified during production. Animal-origin milks contains naturally-occurring vitamins A and trace amounts of vitamins D, E, K, C and B. It is also a source of phosphorus, potassium, zinc and easily digestible magnesium, as well as small amounts of sodium and iron [100].

Regarding special benefits of plant-based beverages, they contain bioactive ingredients with health-promoting effects, such as ß-glucans, phytosterols and polyphenols [12, 101].  Plant-based beverages does not contain lactose or cholesterol [12]. Finally, they are rich in antioxidants, that helps to decrease oxidative stress in the body [12].”

 In consequence, the following references were added to the references list:

Ziarmo, M.; Cichonska, P. Lactic acid bacteria-fermentable cereal- and pseudocereal-based beverages. Microorganisms 2021, 9, 2532.

Cichonska, P.; Ziarno, M. Legumes and legume-based beverages fermented with lactic acid bacteria as a potential carrier of probiotics and prebiotics. Microorganisms, 2022, 10, 91.

Cichońska, P.; Ziebicka, A:, Ziarmo, M. properties of rice-based beverages fermented with lactic acid bacteria and Propionibacterium. Molecules 2022, 27, 2558, doi: 10.3390/molecules27082558.

Mäkinen, O.E.; Wanhalinna, V.; Zannini, E.; Arendt, E.K. Foods for special dietary needs: Non-dairy plant based milk substitutes and fermented dairy type products. Crit. Rev. Food Sci. Nutr. 2016, 56, 339-349.

Walczak, Z.; Florowska, A.; Krygier, K. Plant-based milk beverages-Characteristics and availability in Poland. Food Ind. 2017, 71, 14-18.

Fructuoso, I.; Romano, B.; Han, H.; Raposo, A.; Ariza-Montes, A.; Arya-Castillo, L.; Zandonadi, R.P. An overview of the nutritional aspects of plant-based beverages used as sustitutes for cow´s milk. Nutrients, 2021, 13, 2650.”

Comment 8: Also the impacts of animal-based milk and vegetable beverages on human gut microbiota should be carefully compared and discussed. Please note the composition of animal-based milk and vegetable beverages are very different. The impact of each component in vegetable beverages on gut microbiota should be introduced.

Response 8: Information about the differences in both animal-origin milk and vegetable beverages were added to the revised version of the manuscript:

““Plant-based beverages differ in their composition and nutritional value from animal-origin milks, in terms of its content of proteins, lipids and carbohydrates, and glycemic index [12]. Nutritional value of plant-based beverages varies depending on the raw material from which they are produced, and the production technology employed [12]. Nevertheless, in most cases, plant-based beverages are high in carbohydrates and low in protein, containing up to 30 times less protein that CM. Additionally, protein of plant-based beverages contains lower amount limiting amino-acids (lysine in cereals, methionine in legumes) and poor digestibility than animal-origin milks [12, 98]. Plant-based milk substitutes can be also fermented to obtain diary-free yogurt-type products while rendering the raw material into a more palatable form [99].

In most cases, plant-based beverages are low in fat unless supplemented with vegetable oils. Compared to milk, plant-based beverages have a lower content of saturated fatty acids (SFA), except for coconut beverage, which is SFA-rich. Plant-based beverages are dominated by unsaturated fatty acids, mainly in the form of oleic, linolenic, and linoleic acids [12,98]. Animal-origin milks are natural sources of calcium, and plant-based beverages usually content lower amount of this nutrient if not fortified during production. Animal-origin milks contains naturally-occurring vitamins A and trace amounts of vitamins D, E, K, C and B. It is also a source of phosphorus, potassium, zinc and easily digestible magnesium, as well as small amounts of sodium and iron [100].

Regarding special benefits of plant-based beverages, they contain bioactive ingredients with health-promoting effects, such as ß-glucans, phytosterols and polyphenols [12, 101].  Plant-based beverages does not contain lactose or cholesterol [12]. Finally, they are rich in antioxidants, that helps to decrease oxidative stress in the body [12].”

Regarding discussion about its effects in gut microbiota, it was added new information

Reviewer 2 Report

Comments and Suggestions for Authors

The manuscript is a review article that aims to compare the effects of animal milk and its plant-based alternatives on the human gut microbiota. The authors conducted a comprehensive literature search, collecting and systematizing data from numerous studies on both animal models and human subjects. The work is well documented, with numerous references to scientific publications, attesting to the reliability and solid research background of the authors. The manuscript is written clearly and understandably and meets the standards of academic writing. Some topics, such as the influence of milk processing on the microbiota, are only briefly discussed. It is important to avoid definitive statements, especially since research on the effects of milk and plant-based beverages on the microbiota is still ongoing. In some places there are repetitions in the text that could be improved with a more concise style.

·         In general – Given the changes in the taxonomy of lactic acid bacteria (see: http://lactobacillus.ualberta.ca/), I suggest using the term 'lactobacilli' instead of 'Lactobacillus' in the text and tables, wherever possible.

·         1. Introduction – It might be worth adding a few more sentences about the variety of plant-based drinks available on the market and their growing popularity as an alternative to animal milk. I propose to add a definition of herbal drinks and a brief description of their composition. In its current form, the introduction focuses primarily on the benefits of animal milk. I would suggest a more balanced presentation of both sides of the coin, including the benefits of consuming plant-based drinks. A mention of the influence of diet on the gut microbiota – this could be highlighted in the introduction to emphasize the importance of the topic. It is worth adding a short concluding paragraph at the end of the introduction. It should include a statement of the main aim of the manuscript and its significance in the context of previous research. I would suggest a brief overview of the structure of the manuscript.

·         2. The human gut microbiota – An attempt could be made to discuss the influence of diet on the gut microbiota in more detail. This was only mentioned in a few sentences, but is one of the main themes of the manuscript. I would suggest more emphasis on the importance of short chain fatty acids (SCFAs) for human health.

·         3. Effect of animal-based milk on the gut microbiota – The chapter could delve deeper into several areas: the mechanisms by which individual milk components influence the microbiome; Differences in gut microbiome composition between individuals who regularly consume dairy products and those who do not; and the potential health effects of the observed changes in the microbiome. It would be beneficial to include information about the impact of milk processing on its impact on the microbiome. In addition, a comparison of the effects of milk from animals fed different diets would be of interest. At the end of each section, it would be useful to add a brief summary highlighting the key findings.

·         4. Vegetable beverages and effects on the human gut microbiota –  The text only mentions that these drinks differ in their composition from animal milk. It would be helpful to elaborate on this point and provide detailed information on the protein, fat, carbohydrate, vitamin and mineral content of various plant-based drinks (see: https://doi.org/10.3390/microorganisms10010091; https://doi.org/10.3390/microorganisms9122532). While the chapter describes the general effects of plant-based beverages on the microbiome, a detailed analysis of the effects of individual beverage types is missing. This section should be expanded to reference specific studies and types of gut bacteria.

·         5. Conclusions – In their current form, the conclusions largely summarize information already presented in the previous chapters. Rather than repeating these points, it would be useful to focus on interpreting them and deriving more complex conclusions. The conclusions only superficially compare the effects of animal milk and plant-based beverages on the gut microbiota. This aspect should be expanded by pointing out specific differences in the effects of different types of milk and beverages as well as references to specific studies and bacterial species. It is important to note that knowledge about the effects of milk and plant-based beverages on the gut microbiota is still incomplete. The limitations of previous studies and the need for further analysis should be highlighted. It is important to emphasize that the effects of milk and plant-based drinks on the gut microbiota may vary from person to person. The importance of an individual nutritional approach that takes into account the needs and predispositions of a particular organism should be emphasized.

·         Abstract – The abstract refers to a comparative analysis of animal milk and plant-based beverages on the gut microbiota, but does not describe the precise aspects of this influence that were examined. A more explicit statement of the aim of the study is appropriate. While the summary provides broad conclusions, it lacks concrete examples illustrating the positive effects of animal milk and the potentially negative effects of plant-based alternatives. Finally, a statement highlighting the practical implications of the results, such as their application in dietary guidelines, would be valuable.

Author Response

Reviewer 2:

Comments 1: The manuscript is a review article that aims to compare the effects of animal milk and its plant-based alternatives on the human gut microbiota. The authors conducted a comprehensive literature search, collecting and systematizing data from numerous studies on both animal models and human subjects. The work is well documented, with numerous references to scientific publications, attesting to the reliability and solid research background of the authors. The manuscript is written clearly and understandably and meets the standards of academic writing. Some topics, such as the influence of milk processing on the microbiota, are only briefly discussed.

Response 1: The authors sincerely thank the constructive words from the Reviewer. It was included more information about the milk processing effects on the microbiota. In concrete:

“It should be considered that animal milks cannot always be consumed in its natural form, and to extend its shelf life, as well as to ensure its microbial safety, it often requires processing, including refrigeration, homogenization and heat treatment [58]. The most common heat treatments widely used in the dairy industry to achieve milk safety and preservation are pasteurization and ultra-high temperature (UHT) sterilization. Thermi-zation and in-bottle sterilization are also performed on raw milk [59]. However, the heat treatments applied to milk not only ensure its microbiological safety, but also modify the organoleptic properties and composition of the milk [59]. These effects on nutritional composition include denaturation of some protein fractions as whey proteins and en-zymes, lactose degradation, and inactivation of potentially functional components [60].

Both refrigeration, homogenization and thermal treatments can also affect milk mi-crobiota [59]. Refrigeration time favors significant changes in the quantity and population composition of microorganisms in raw milk, increasing genera as Xanthobacterium, Pseu-domonas, and Lactococcus [61]. Pasteurization and UHT destroy bacteria, toxin-producing and spore-forming organisms, and UHT destroys all vegetative microorganisms [59]. However, the thermal treatments also destroy lactic acid bacteria that are commonly in raw milk, such as lactobacilli or Lachnospiraceae [62].”

Consequently, the following references were added to the references list:

Melini, F.; Melini V.; Luziatelli, F.; Ruzzi, M. Raw and heat-treated milk: From public health risks to nutritional qual-ity. Beverages 2017, 3, 54.

Felfoul, I., Beaucher, E.; Cauty, C.; Attia, H.; Gaucheron, F.; Ayadi, M. Deposit generation during camel and cow milk heating: microstructure and chemical composition. Food Bioprocess Technol. 2016, 9, 1268-1275.

Zhang, Y.; Yu, P.; Tao, F. Dynamic interplay between microbiota shifts and differential metabolites during dairy processing and storage. Molecules 2024, 29, 2745.

Perdomo, A.; Calle, A. Assessment of microbial communities in a dairy farm from a food safety perspective. Int. J. Food Microbiol. 2024, 423, 110827.

Comments 2: It is important to avoid definitive statements, especially since research on the effects of milk and plant-based beverages on the microbiota is still ongoing. In some places there are repetitions in the text that could be improved with a more concise style.

Response 2: Thank you for your comments. The authors checked the manuscript and avoided definitive statements. Additionally, to make the text more concise, the following repetitions were avoided:

Line 81: “there are some concerns” was changed to “there may be some disadvantages”.

Line 110: “In addition” was deleted.

Line 151: “Lactose is one of the components that often affects the consumption of milk, since, globally, approximately 70% of the adult population in different countries is unable to digest lactose, the sugar present in milk, completely” was deleted.
line 214: “Little is known about the effects of milk fat on the GM composition” was deleted.

Line 221: “natural trilayered” was deleted.

Line 284: “(C8:0)” was deleted.

Line 291: “have also been reported” was deleted.

Line 335: “The Bifidobacterium genus is considered beneficial because it is correlated with many positive health outcomes and is negatively correlated with obesity and weight gain [27].” Was deleted.

Line 338: “This in vitro study revealed that mare milk is a very adequate substitute for human milk [27].” was deleted.

Line 377: “Lactoferrin has been shown to have antibacterial activity against Helicobacter species [75].” was deleted.

Line 412: “and make CAM more suitable for individuals who are lactose intolerant” was deleted.

Line 432: “, the production of SCFAs” was deleted.

Line 433: “CAM intake was also reported to increase the GM abundances of Lachnospiraceae and Muribaculaceae” was deleted.

Line 437: “The acetic, propionic and butyric acid levels of mice that were administered CAM in-creased significantly [81].” Was deleted.

Line 445: “Desulfovibrio,” was eleted.

Paragraph in lines 448-457 was moved to previously to “In contrast, CAM intake…”.

Line 466: Contain” was changed to “can contain”.

Line 472: “although they are well known to” was changed to “although vegetable beverages can”

Line 473: “contain few nutrients” was changed to “can contain fewer nutrients than dairy milks”

Line 479: “has several” was changed to “were reported to provide some”

Line 482: “there are some concerns” was changed to “it was reported that could be some concerns”

Lines 484-486: “, and there are some concerns regarding plant-based formulas and their possible effects on sexual development and reproduction, neurobehavioral development, immune function, and thyroid function” because it was reiterative.

Line 494: “are” was changed to “can be considered”

Line 498: “that these methods are not beneficial to the subject” was changed to “some results cannot be considered as beneficial to the host”.

Comments 3: In general – Given the changes in the taxonomy of lactic acid bacteria (see: http://lactobacillus.ualberta.ca/), I suggest using the term 'lactobacilli' instead of 'Lactobacillus' in the text and tables, wherever possible.

Response 3: According to the suggestions from the Reviewer, the term “lactobacillus” was changed to “lactobacilli” in all the cases in that a species name was included.

Comments 4: Introduction – It might be worth adding a few more sentences about the variety of plant-based drinks available on the market and their growing popularity as an alternative to animal milk. I propose to add a definition of herbal drinks and a brief description of their composition. In its current form, the introduction focuses primarily on the benefits of animal milk.

Response 4: Thank you very much. You are right, the Introduction sections was very focused in animal milks. In the revised form, the following paragraphs were added:

 “Plant-based substitutes can be defined as an emulsion that resembles animal-origin milk in consistency and appearance, that are made following a general procedure that in-cludes the aqueous extraction of the plant material, removal of remaining soil parts, and afterwards a thermal treatment of the fluid [12]. The most employed vegetable sourced to make plant-based beverages can be legumes, cereals, pseudocereals, seeds or nuts, oilseeds plants or even potatoes [13]. Nowadays, the most popular plant-based beverages are soy, almond, coconut, oat and rice [13].”

Comments 5: I would suggest a more balanced presentation of both sides of the coin, including the benefits of consuming plant-based drinks. A mention of the influence of diet on the gut microbiota – this could be highlighted in the introduction to emphasize the importance of the topic. It is worth adding a short concluding paragraph at the end of the introduction. It should include a statement of the main aim of the manuscript and its significance in the context of previous research. I would suggest a brief overview of the structure of the manuscript.

Response 5: Thank you very much for your comment. A new mention of diet in gut microbiota was included, although according to the instruction of other reviewer, it was added in the “ 2. The human gut microbiota” subheading. In concrete, it was added the following information:

“The composition of the healthy gut microbiota is dominated for up to 90% by the phyla Firmicutes and Bacteroidetes [5]. The phylum of Firmicutes includes several genera, of which the most common (up to 95%) are Lactobacillus, Bacillus, Enterococcus, Ruminicoccus and Clostridium [31]. GM plays key roles for human health, including a shield effect with protection of the gut barrier, shaping and maturation of the immune system, the regula-tion of human metabolism and nutrients and drug absorption [31].

GM is a highly dynamic ecosystem influenced by numerous factors such as age, ge-netics, environment, lifestyle and, most prominently, diet. While different compositions of diet can lead to changes of microbiota profiles, an unbalanced diet consumed over a long period, as Western Diet [32] accompanied by an unhealthy lifestyle can leave traces of ev-idence on the GM as well. Generally, subjects with obesity and overweight tend to show a lower diversity of GM [32].”

At the end of the introduction, it was moved as concluding paragraph the following paragraph:

“However, despite its common consumption, the number of articles that have investigated the effects of animal milk and its natural vegetable substitutes on the GM is relatively low. Thus, the aim of this narrative literature search was conducted up to June 2024 for all the available literature in the Web of Science and Scopus. A combination of the following search terms was applied: “milk”; “milk substitute”; “vegetal beverages”; “plant-based milk” as the topic; and “gut microbiota” in the title. A total of 64 articles were ultimately selected and included in the review, as were the other 48 articles that were included to contextualize the article.”

According to the shot of the manuscript structure, despite a lot of new information was added in response to the reviewers´ suggestions, the manuscript was even increased. However, as can be seem in the response to the Comments 2, several parts of the text that were considered redundant have been eliminated to make the manuscript easier to read

Comment 6: The human gut microbiota – An attempt could be made to discuss the influence of diet on the gut microbiota in more detail. This was only mentioned in a few sentences but is one of the main themes of the manuscript. I would suggest more emphasis on the importance of short chain fatty acids (SCFAs) for human health.

Response 6: According to the suggestions from the reviewer, bot new information about effect of diet on gut microbiota and the effects of short-chain fatty acids were added to the revised version of the manuscript.

Regarding gut microbiota and diet effect in it, it was added the following paragraph: “The composition of the healthy gut microbiota is dominated for up to 90% by the phyla Firmicutes and Bacteroidetes [5]. The phylum of Firmicutes includes several genera, of which the most common (up to 95%) are Lactobacillus, Bacillus, Enterococcus, Ruminicoccus and Clostridium [31]. GM plays key roles for human health, including a shield effect with protection of the gut barrier, shaping and maturation of the immune system, the regula-tion of human metabolism and nutrients and drug absorption [31].

Regarding effects of short chain fatty acids, it was added the following paragraph: “), that can be defined as the primary end products formed by non-digestible carbohydrates (in some cases also peptides and proteins) when fermented by GM [Rehka et al., 2024]. The most important SCFAs are acetate, propionate and butyrate [13], that are easily soluble and absorbed quickly in >95% total, being the remaining excreted in feces [Rehka et al., 2024]. To produce each of them a different mechanism and fermenting bacteria are needed. Acetate is mainly produced from pyruvate via acetyl CoA, requiring the participation of GM bacterial genus such as Akkermansia, Bacteroides or Bifidobacterium [Rehka et al., 2024]. Propionate is formed from different synthesis ways: it can be produced from lactate (participating Megaspharea elsdenii and Coprococcus catus), from succinate (participating Bacteroides, Dialister or Roseburia genera) or from fucose and rhamnose (participating Salmonella and Roseburia genus) [Reichardt et al., 2014]. SCFAs are absorbed by colonocytes (mainly butyrate, a vital energy source for colocytes) and effluxed into the blood [Rehka et al., 2024]. SCFAs can also act as signalling molecules by activating G-protein-coupled receptors and inhibiting histone deacetylases    [Rehka et al., 2024]”

Accordingly, the following references were added to reinforce the new information:

Rinniella, E.; Tohumcu, E.; Raoul, P.; Fiorani, M.; Cintoni, M.; Mele, M.C.; Cammarota, G.; Gasbarrini, A. The role of diet shaping human gut microbiota. Best Pract. Res. Clin. Gastroenterol. 2023, 62-63, 101828.

Perler, B.K.; Friedman, E.S.; Wu, G.D. The role of the gut microbiota in the relationship between diet and human health. Annu. Rev. Physiol. 2023, 85, 449-469.

Rekha, K.; Venkidasamy, B.; Samynathan, R.; Nagella, P.; Rebezov, M.; Khayrullin, M.; Ponomarev, E.; Bouyaha, A.; Sarkar, T.; Shariati, M.A.; Thiruvengadam, M.; Simal-Gandara, J. Short-chain fatty acid: An updated review on signaling, metabolism, and therapeutic effects. Crit. Rev. Food Sci. Nutr. 2024, 64(9), 2461-2489.

Reichardt, N.; Duncan, S.H.; Young, P.; Belenger, A.; McWilliam Leitch, C.; Scott, K.P.; Flint, H.J.; Lousi, P. Phylog-netic distribution of three pathways for propionate production within the human gut microbiota. ISME J. 2014, 8(6), 1323-1335.

Comment 7: Effect of animal-based milk on the gut microbiota – The chapter could delve deeper into several areas: the mechanisms by which individual milk components influence the microbiome; Differences in gut microbiome composition between individuals who regularly consume dairy products and those who do not; and the potential health effects of the observed changes in the microbiome. It would be beneficial to include information about the impact of milk processing on its impact on the microbiome. In addition, a comparison of the effects of milk from animals fed different diets would be of interest. At the end of each section, it would be useful to add a brief summary highlighting the key findings.

Response 7: With respect to the mechanisms by which individual milk components influence the microbiome it was included a new table (Table 1)

Table 1 comprised a brief description of the effects reported about the impact of different milk components on GM.

Table 1. Effects of animal milk components on human gut microbiota.

Milk component

Effects on gut microbiota

References

Lactose

Increase lactobacilli and Bifidobacterium growth

[7,27]

Milk oligosaccharides

Favors growth of beneficial bacteria as lactobacilli and Bifidobacterium; Inhibiting bacterial adhesion of pathogens to enterocytes

[12,28,59]

α-lactalbumin

Promoting growth of beneficial bacteria and exerts antimicrobial activity against some pathogens

[12]

Lactoferrin

Prebiotic effect and inhibition of pathogens

[12,35,43,44]

Lysozyme

Increase resistance to intestinal colonization by some pathogens

[39]

Milk fat globule membrane

Promote the formation of binding groups on the surface of probiotics; In vitro bactericidal effects against some pathogens

[28,41,49,50]

With respects to include information about differences in gut microbiome composition between individuals who regularly consume dairy products and those who do not; and the potential health effects of the observed changes in the microbiome, it was included:

“A recent review [Aslam et al. 2020] showed that milk and dairy products intake can modulate GM in a beneficial manner. Dairy foods appeared to facilitate the growth of beneficial microbes as lactobacilli and Bifidobacterium. However, due to the heterogeneity in study methods and outcome reporting, it is difficult to draw robust conclusions.”

With respects to include information about differences milk from animals fed different diets would be of interest, it was added the following paragraph: “It should be considered that feeding using in livestock widely influences the composition of the milk, not only in proximate composition but also in minor compounds such as terpenes, phytanic acid, pristanic acid, skatole, antioxidants of fatty acids [Vicente et al., 2017].”

Accordingly, the following reference was added:

Vicente, F.; Santiago, C.; Jiménez-Calderón, J.D.; Martínez-Fernández, A. Capacity of milk composition to identify the feeding system used to dairy cows. J. Dairy Res. 2017, 84, 254-263.

With respect to the effect about the impact of milk processing on its impact on the microbiome, it was included the paragraphs:

“It should be considered that animal milks cannot always be consumed in its natural form, and to extend its shelf life, as well as to ensure its microbial safety, it often requires processing, including refrigeration, homogenization and heat treatment [58]. The most common heat treatments widely used in the dairy industry to achieve milk safety and preservation are pasteurization and ultra-high temperature (UHT) sterilization. Thermi-zation and in-bottle sterilization are also performed on raw milk [59]. However, the heat treatments applied to milk not only ensure its microbiological safety, but also modify the organoleptic properties and composition of the milk [59]. These effects on nutritional composition include denaturation of some protein fractions as whey proteins and en-zymes, lactose degradation, and inactivation of potentially functional components [60].

Both refrigeration, homogenization and thermal treatments can also affect milk mi-crobiota [59]. Refrigeration time favors significant changes in the quantity and population composition of microorganisms in raw milk, increasing genera as Xanthobacterium, Pseu-domonas, and Lactococcus [61]. Pasteurization and UHT destroy bacteria, toxin-producing and spore-forming organisms, and UHT destroys all vegetative microorganisms [59]. However, the thermal treatments also destroy lactic acid bacteria that are commonly in raw milk, such as lactobacilli or Lachnospiraceae [62].

Additionally, it should be noted that there are very important differences in the com-position, structure and physicochemical properties of milk depending on the animal spe-cies from which it originates. Thus, in the case of protein content, although there are rele-vant differences in total protein content among CM and other milks such as goat, sheep of camel milk (CAM), milks from these latter species have a lower casein-to-whey-protein ra-tio as well as a relatively higher β-casein ratio-to-α-casein ratio compared to CM [63]. Fat content varies significantly even within the same animal species depending on factors such as breed (3.3-5.4% in CM, 3-7.2% in goat milk (GOM) and 2-6% in CAM) [63]. GM is richer in SCFAs and medium chain triacylglycerols than CM, whereas CAM contains a higher rate of long chain fatty acids and a lower rate of SCFAs than CM [64]. With respect to lactose content, CM contains higher content (4.4-5.6%) than GOM (3.2-5%) or CAM (3.5-5.1%) [63].

It should be considered that feeding using in livestock widely influences the compo-sition of the milk, not only in proximate composition but also in minor compounds such as terpenes, phytanic acid, pristanic acid, skatole, antioxidants of fatty acids [65].

3.1. Effects of cow milk on the gut microbiota

Comments 8. Vegetable beverages and effects on the human gut microbiota –  The text only mentions that these drinks differ in their composition from animal milk. It would be helpful to elaborate on this point and provide detailed information on the protein, fat, carbohydrate, vitamin and mineral content of various plant-based drinks (see: https://doi.org/10.3390/microorganisms10010091; https://doi.org/10.3390/microorganisms9122532).

Response 8: Thank you very much for your comment. In the revised form of the manuscript, the following information regarding the composition of plant-based beverages and its differences with animal-origin milks was added:

“Plant-based substitutes cab be defined as an emulsion that resembles animal-origin milk in consistency and appearance, that are made following a general procedure that includes the aqueous extraction of the plant material, removal of remaining soil parts, and afterwards a thermal treatment of the fluid [Ziarmo, 2021]. After this procedure, the obtained beverages differ in their composition and nutritional value from animal-origin milks, in terms of its content of proteins, lipids and carbohydrates, and glycemic index []. Plant-based beverages differ in their composition and nutritional value from animal-origin milks, in terms of its content of proteins, lipids and carbohydrates, and glycemic index [12]. Nutritional value of plant-based beverages varies depending on the raw material from which they are produced, and the production technology employed [12]. Nevertheless, in most cases, plant-based beverages are high in carbohydrates and low in protein, containing up to 30 times less protein that CM. Additionally, protein of plant-based beverages contains lower amount limiting amino-acids (lysine in cereals, methionine in legumes) and poor digestibility than animal-origin milks [12, 98]. Plant-based milk substitutes can be also fermented to obtain diary-free yogurt-type products while rendering the raw material into a more palatable form [99].

In most cases, plant-based beverages are low in fat unless supplemented with vegetable oils. Compared to milk, plant-based beverages have a lower content of saturated fatty acids (SFA), except for coconut beverage, which is SFA-rich. Plant-based beverages are dominated by unsaturated fatty acids, mainly in the form of oleic, linolenic, and linoleic acids [12,98]. Animal-origin milks are natural sources of calcium, and plant-based beverages usually content lower amount of this nutrient if not fortified during production. Animal-origin milks contain naturally-occurring vitamins A and trace amounts of vitamins D, E, K, C and B. It is also a source of phosphorus, potassium, zinc and easily digestible magnesium, as well as small amounts of sodium and iron [100].

Regarding special benefits of plant-based beverages, they contain bioactive ingredients with health-promoting effects, such as ß-glucans, phytosterols and polyphenols [12, 101].  Plant-based beverages does not contain lactose or cholesterol [12]. Finally, they are rich in antioxidants, that helps to decrease oxidative stress in the body [12].

In consequence, the following references were added to the references list:

Ziarmo, M.; Cichonska, P. Lactic acid bacteria-fermentable cereal- and pseudocereal-based beverages. Microorganisms 2021, 9, 2532.

Cichońska, P.; Ziarno, M. Legumes and legume-based beverages fermented with lactic acid bacteria as a potential carrier of probiotics and prebiotics. Microorganisms, 2022, 10, 91.

Cichońska, P.; Ziebicka, A., Ziarmo, M. properties of rice-based beverages fermented with lactic acid bacteria and Propionibacterium. Molecules 2022, 27, 2558, doi: 10.3390/molecules27082558.

Mäkinen, O.E.; Wanhalinna, V.; Zannini, E.; Arendt, E.K. Foods for special dietary needs: Non-dairy plant based milk substitutes and fermented dairy type products. Crit. Rev. Food Sci. Nutr. 2016, 56, 339-349.

Walczak, Z.; Florowska, A.; Krygier, K. Plant-based milk beverages-Characteristics and availability in Poland. Food Ind. 2017, 71, 14-18.

Fructuoso, I.; Romano, B.; Han, H.; Raposo, A.; Ariza-Montes, A.; Arya-Castillo, L.; Zandonadi, R.P. An overview of the nutritional aspects of plant-based beverages used as sustitutes for cow´s milk. Nutrients, 2021, 13, 2650.”

Comment 9: While the chapter describes the general effects of plant-based beverages on the microbiome, a detailed analysis of the effects of individual beverage types is missing. This section should be expanded to reference specific studies and types of gut bacteria.

Response 9: Thank you for your comment. New information has been included in the modified version. There are actually many studies that study the effects on the intestinal microbiota of fermented vegetable beverages to which probiotics have been added, but few that have studied the effects of ingestion of the vegetable beverage without added probiotics. Even so, the information regarding plant-based beverages has been increased in the revised version of the manuscript to approximately double that present before the revision, by the addition of the following paragraphs:

“Plant-based beverages differ in their composition and nutritional value from animal-origin milks, in terms of its content of proteins, lipids and carbohydrates, and glycemic index [12]. Nutritional value of plant-based beverages varies depending on the raw material from which they are produced, and the production technology employed [12]. Nevertheless, in most cases, plant-based beverages are high in carbohydrates and low in protein, containing up to 30 times less protein that CM. Additionally, protein of plant-based beverages contains lower amount limiting amino-acids (lysine in cereals, methionine in legumes) and poor digestibility than animal-origin milks [12, 98]. Plant-based milk substitutes can be also fermented to obtain diary-free yogurt-type products while rendering the raw material into a more palatable form [99].

In most cases, plant-based beverages are low in fat unless supplemented with vegetable oils. Compared to milk, plant-based beverages have a lower content of saturated fatty acids (SFA), except for coconut beverage, which is SFA-rich. Plant-based beverages are dominated by unsaturated fatty acids, mainly in the form of oleic, linolenic, and linoleic acids [12,98]. Animal-origin milks are natural sources of calcium, and plant-based beverages usually content lower amount of this nutrient if not fortified during production. Animal-origin milks contain naturally-occurring vitamins A and trace amounts of vitamins D, E, K, C and B. It is also a source of phosphorus, potassium, zinc and easily digestible magnesium, as well as small amounts of sodium and iron [100].

Regarding special benefits of plant-based beverages, they contain bioactive ingredients with health-promoting effects, such as ß-glucans, phytosterols and polyphenols [12, 101].  Plant-based beverages does not contain lactose or cholesterol [12]. Finally, they are rich in antioxidants, that helps to decrease oxidative stress in the body [12].”

“The high activity of lactic acid bacteria during the lactic acid fermentation of vegetable beverages causes similar changes in the product composition to that observed in lactic acid bacteria-fermented milk, including the formation of organic acids, acidification of the environment, decomposition of some carbohydrates, and digestion of proteins and lipids. Thus, fermented vegetal beverages can be an alternative to fermented milk to meet the growing demand for this type of product among consumers [102].

“Comparing individually the results obtained for specific vegetable beverages, it was found that after almond milk consumption achieved in a rat model a higher relative abundance in Bacteroidetes (Bacteroidaceae) and Firmicutes populations (Lactobacillaceae, Clostridiaceae and Peptostreptococcaceae) than soymilk of CM consumption [74]. Bacteroidetes members have gene-encoded carbohydrate active enzymes than can switch readily between different energy sources in the gut, depending on availability, whereas some genus of Firmicutes were described and SFCAs producers [74]. Interestingly, the results obtained by the same work regarding bone density prevention showed similar results in both CM, soy beverage or almond beverage [74]. This result is important because it suggests that vegetable drinks, despite its lower content of calcium than animal-origin milks, can play a positive role in maintaining bone density.

Regarding soy beverages, most of the results obtained about its effects on GM were beneficial, including increases in lactobacilli [11,74,107] and Bifidobacterium [108,109,113], or decreased in Proteobacteria [77,108]. Most of results were beneficial, with the few exceptions of increases in Salmonella enterica or Fusobacterium [11]. In both cases, the authors of the work claim that their increases have been in low proportions and that they still represented a minor percentage of the bacterial groups present in the GM [11].

Comment 10: Conclusions – In their current form, the conclusions largely summarize information already presented in the previous chapters. Rather than repeating these points, it would be useful to focus on interpreting them and deriving more complex conclusions. The conclusions only superficially compare the effects of animal milk and plant-based beverages on the gut microbiota. This aspect should be expanded by pointing out specific differences in the effects of different types of milk and beverages as well as references to specific studies and bacterial species. It is important to note that knowledge about the effects of milk and plant-based beverages on the gut microbiota is still incomplete. The limitations of previous studies and the need for further analysis should be highlighted. It is important to emphasize that the effects of milk and plant-based drinks on the gut microbiota may vary from person to person. The importance of an individual nutritional approach that takes into account the needs and predispositions of a particular organism should be emphasized.

Response 10: Thank you very much for your comment. All your points are interesting and were incorporated to the new version of the conclusions. Thus, most of the conclusions section was changed to:

“ The consumption of vegetable milk substitutes is increasing worldwide, especially in some geographical areas, such as Europe.

However, in addition to the macronutrient and calcium contents, which plant-based beverages mimic, milk contains numerous minor compounds, a unique structural com-position of fats and a mixture of protein fractions that can affect the human GM. There are important differences in protein content and composition, lipid composition, absence of lactose and cholesterol, and different amounts of minor components. It should be also considered that the composition of animal milk varies greatly depending on factors such as the species and breed of animal from which it comes, the animal's feed and the indus-trial treatment it undergoes. Vegetable beverages, on the other hand, have a more stable composition, since their initial composition and technological treatment is usually stable.

With respect to the effects of both types of beverages on the intestinal microbiota, the first conclusion that can be drawn from this work is that there is little information on the subject. Few studies have tested the effects of animal milk or plant-based beverages per se (without added prebiotics or probiotics) on the GM, especially in humans. Thus, knowledge about the effects of milk and plant-based beverages on the GM is still incom-plete and much more work is needed before an adequate consensus can be established.

Based on the results shown so far, it appears that the consumption of milk of animal origin exerts beneficial effects on human GM. In contrast, vegetable beverages also show mostly positive results, but in some cases also favoring the growth of potentially negative bacterial genera. Therefore, at least from the point of view of their effects on the GM, it cannot be said that plant-based milk replacers are a perfect substitute for milk, the latter being generally more beneficial for the composition of the human GM.

All this being considered with the due limitations, since these are in vitro studies, in animals or with few human subjects. The effects of milk and plant-based drinks on the GM may vary from person to person, and and therefore, before recommending the popu-lation to opt for milk of animal origin or its vegetable substitutes, it is very important to adopt an individual nutritional approach that takes into account the needs and predispo-sitions of each consumer.

Comment 11: Abstract – The abstract refers to a comparative analysis of animal milk and plant-based beverages on the gut microbiota, but does not describe the precise aspects of this influence that were examined. A more explicit statement of the aim of the study is appropriate. While the summary provides broad conclusions, it lacks concrete examples illustrating the positive effects of animal milk and the potentially negative effects of plant-based alternatives. Finally, a statement highlighting the practical implications of the results, such as their application in dietary guidelines, would be valuable.

Response 11: Thank you for your comments. In the revised version of the manuscript, some introductory information was deleted from the abstract, and it were included the following supplementary information.

Describe the precise aspects of this influence that were examined : “The aim of this review is to compare the effects obtained by previous works on the composition of the gut microbiota after the ingestion of animal milk and/or vegetable beverages”.

More specific information about animal-origin milks was included: “Thus, it was found an increase of gut microbiota richness and diversity, increase in the production of short chain fatty acids, and beneficial microbes such as Bifidobacterium, lactobacilli, Akkermansia, Lachnospiraceae or Blautia. In other cases, significant decrease in potential harmful bacteria as Proteobacteria, Erysipelotrichaceae, Desulfovibrionaceae or Clostridium perfingens were found after animal-origin milk intake.”

More specific information about vegetable substitutes on gut microbiota: “Vegetable beverages have also generally produced positive results in gut microbiota as the increase in the relative presence of lactobacilli, Bifidobacterium or Blautia. However, it was also found some potential negative results such as increases in the presence of potential pathogens such as…”

A statement highlighting the practical implications of the results, such as their application in dietary guidelines, would be valuable: it was added “These different effects on the intestinal microbiota should be considered in those cases where the replacement of animal milks by vegetable substitutes is recommended.”

Round 2

Reviewer 1 Report

Comments and Suggestions for Authors

The revised manuscript can be accepted.

Comments on the Quality of English Language

Minor editing of English language required.

Reviewer 2 Report

Comments and Suggestions for Authors

The author responded to all the queries. I do not have any questions.